# Glacial $CO_2$ decrease and deep-water deoxygenation by iron fertilization from glaciogenic dust

Akitomo Yamamoto[1,2], Ayako Abe-Ouchi[1,2], Rumi Ohgaito[1], Akinori Ito[1], Akira Oka[2]

[1]Japan Agency for Marine-Earth Science and Technology, Yokohama, Japan
[2]Atmospheric and Ocean Research Institute, The University of Tokyo, Kashiwa, Japan

*Corresponding author*: A. Yamamoto (akitomo@jamstec.go.jp)

**Abstract**

**Increased accumulation of respired carbon in the deep ocean associated with enhanced efficiency of the biological carbon pump is thought to be a key mechanism of glacial $CO_2$ drawdown. Despite greater oxygen solubility due to seawater cooling, recent quantitative and qualitative proxy data show glacial deep-water deoxygenation, reflecting increased respired carbon accumulation. However, the mechanisms of deep-water deoxygenation and contribution from the biological pump to glacial $CO_2$ drawdown have remained unclear. In this study, we report the significance of iron fertilization from glaciogenic dust in glacial $CO_2$ decrease and deep-water deoxygenation using our numerical simulation, which successfully reproduces the magnitude and large-scale pattern of the observed oxygen changes from the present to the Last Glacial Maximum. Sensitivity experiments show that physical changes contribute to only one-half of all glacial deep deoxygenation whereas the other one-half is driven by iron fertilization and an increase in the whole ocean nutrient inventory. We find that iron input from glaciogenic dust with higher iron solubility is the most significant factor in enhancing the biological pump and deep-water deoxygenation. Glacial deep-water deoxygenation expands the hypoxic waters in the deep Pacific and Indian oceans. The simulated global volume of hypoxic waters is nearly double the present value, suggesting that glacial deep-water was a more severe environment for benthic animals than that of the modern oceans. Our model underestimates the deoxygenation in the deep Southern Ocean because of enhanced ventilation. The model-proxy comparison of oxygen change suggests that a stratified Southern Ocean is required for reproducing the oxygen decrease in the deep Southern Ocean. Iron fertilization and a global nutrient increase contribute to a decrease in glacial $CO_2$ of more than 30 ppm, which is supported by the model-proxy agreement**

**of oxygen change. Our findings confirm the significance of the biological pump in glacial CO$_2$ drawdown and**

**deoxygenation.**

# 1 Introduction

The oceanic carbon cycle has been proposed as a driver of glacial–interglacial $CO_2$ change; however, the magnitude of glacial $CO_2$ reduction of 80-100 ppm has yet to be fully reproduced by numerical model simulations using both an ocean general circulation model (OGCM) and a biogeochemical model (Ciais et al., 2013). The oceanic soft-tissue biological pump, by which the photosynthetic production, sinking, and remineralization of organic matter store dissolved inorganic carbon in the deep ocean, is among the mechanisms controlling glacial-interglacial as well as future atmospheric $CO_2$ change (Sarmiento and Gruber 2006; Sigman et al., 2010; Yamamoto et al., 2018). During glacial periods, the efficiency of the biological pump would have been enhanced by biogeochemical processes (e.g. dust-borne iron fertilization (Martin, 1990) and an increase in nutrient inventory associated with a sea-level drop (Broecker, 1982; Wallmann et al., 2016)), leading to the transfer of carbon from the atmosphere to the deep ocean. Although changes in marine productivity during glacial periods and its relationship to the dust deposition flux have been widely supported by proxy records (Kohfeld et al., 2005; Jaccard et al., 2013), there are no direct proxy records of the greater accumulation of respired organic carbon. Thus, the contribution of the biological pump to glacial $CO_2$ reduction is poorly understood.

Because the dissolved oxygen cycle is the mirror image of the biological carbon cycle (oxygen is produced by photosynthesis and is utilized with consistent stoichiometry through the remineralization of sinking organic matter in the ocean interior), oxygen is consumed in the ocean interior when respired organic carbon accumulates in seawater. Thus, reconstructed oxygen change is useful to constrain the biological pump magnitude and respired carbon accumulation. Proxy data show that, despite greater oxygen solubility due to lower sea surface temperatures (SSTs), oxygen concentrations decreased throughout the deep ocean during the Last Glacial Maximum (LGM) (Jaccard and Galbraith, 2012). This indicates greater oxygen consumption and respired carbon accumulation, which could have been caused by several processes including greater organic matter transport into the deep ocean, increasingly restricted air-sea exchange due to sea-ice expansion, and/or more sluggish ocean circulation. However, previous modeling studies have shown conflicting oxygen changes in LGM simulations (Galbraith and Jaccard, 2015; Schmittner and Somes, 2016; Buchanan et al., 2016; Bopp et al., 2017; Somes et al., 2017; Galbraith and de Lavergne, 2018) and the causes of the oxygen decrease in the deep ocean have not yet been fully explored.

Furthermore, because most observations provide only qualitative estimates of oxygen changes, previous model-proxy comparisons have only discussed the glacial oxygen trend (oxygenation in the upper ocean and deoxygenation in the deep ocean). Several recent studies using $\delta^{13}C$ in benthic foraminiferal or iodine-to-calcium ratios in planktonic foraminifera were able to quantify oxygen concentration changes (Schmiedl and Mackensen, 2006; Hoogakker et al., 2015, 2018; Gottschalk et al., 2016; Lu et al., 2016; Bunzel et al., 2017; Umling and Thunell, 2018). These quantitative proxy data provide firmer constraints on respired carbon accumulation, such that a quantitative model-proxy comparison of oxygen change is very useful for quantifying the contribution of the biological pump to glacial $CO_2$ drawdown.

In this study, to quantify the impact of changes in the biological pump on glacial carbon and oxygen cycles, we conducted pre-industrial (PI) and LGM simulations using the coupled atmosphere–ocean general circulation model (Oka et al., 2011), aerosol model (Ohgaito et al., 2018), and ocean biogeochemical model (Yamamoto et al., 2015). We focused here on the iron fertilization process in enhancing the biological pump. We attempted to separately quantify iron fertilization effects from desert dust and glaciogenic dust (derived from glacier erosion). Previous studies using mineral aerosol models suggest that glaciogenic dust significantly contributed to an increase in the dust deposition flux at high latitudes during the LGM (e.g. the glaciogenic dust derived from Patagonian glaciers increased dust deposition in the Southern Ocean (SO)) and provided a LGM dust deposition flux distribution more consistent with the reported measurements (Mahowald et al., 2006; Ohgaito et al., 2018). Moreover, the iron solubility in glaciogenic dust (~3%) is much higher than that in desert dust (~1%) (Schroth et al., 2009); however, the higher solubility effect of glaciogenic dust on iron fertilization was not considered in previous modeling studies. Glaciogenic dust is a significant source of bioavailable iron (Shoenfelt et al., 2018) and would therefore have a major impact on biological productivity in high nutrient and low chlorophyll (HNLC) regions where biological productivity is limited by the lack of iron. We also considered the effect of an increase in macronutrients inventory associated with a glacial sea level drop of ~120 m (Broecker, 1982; Wallmann et al., 2016). A decrease in the area of continental margins reduced the burial of organic matter in margin sediments, leading to increases in the global inventory of phosphate ($PO_4$) and nitrate ($NO_3$). Based on a recent simulation, increases in $NO_3$ and $PO_4$ inventories by 15% can be assumed (Wallmann et al., 2016).

We performed several sensitivity experiments as listed in Table 1 to explore the contribution of changes in atmospheric dust
and nutrient inventory on glacial carbon and oxygen cycles. Moreover, our modeled oxygen changes were compared to recently
reported qualitative (Jaccard and Galbraith, 2012) and quantitative reconstructions (Schmiedl and Mackensen, 2006;
Hoogakker et al., 2015, 2018; Gottschalk et al., 2016; Lu et al., 2016; Bunzel et al., 2017; Umling and Thunell, 2018) to
evaluate the simulated accumulation of respired carbon. Our simulation shows that glaciogenic dust and increased nutrient
inventory play a crucial role in glacial $CO_2$ decrease and deep-water deoxygenation.

**2 Model and experiments**
The ocean biogeochemical cycle was calculated using the Model for Interdisciplinary Research on Climate (MIROC)-based
offline biogeochemical model, based on Yamamoto et al. (2015), with the implementation of an iron cycle. A one box
atmosphere is coupled to an offline biogeochemical model to predict atmospheric $CO_2$ concentration through gas exchange
between the atmosphere and ocean surface. For the tracer calculation, the model uses prescribed monthly output data of
horizontal ocean velocities, vertical diffusivity, temperature, salinity, sea surface height, sea surface wind speed, sea-ice
fraction, and sea surface solar radiation derived from PI and LGM simulations conducted by Oka et al. (2011) using the MIROC
4m AOGCM. Both PI and LGM simulations follow the PMIP2 protocol (Braconnot et al., 2007). MIROC 4m simulates the
weaker and shallower Atlantic Meridional Overturning Circulation (AMOC) during the LGM (see Fig. 1 in Oka et al. (2011)),
which is consistent with $\delta^{13}C$ distributions reported from proxy data (Curry and Oppo, 2005). The horizontal and vertical
resolutions of the offline biogeochemical model are the same as those in MIROC 4m.

This biogeochemical model includes two phytoplankton classes (nitrogen fixers and other phytoplankton), zooplankton,
particulate detritus, nitrate ($NO_3$), phosphate ($PO_4$), dissolved iron (DFe), dissolved oxygen ($O_2$), dissolved inorganic carbon
(DIC), alkalinity (ALK), two carbon isotopes ($^{13}C$ and $^{14}C$), and an ideal age tracer. The ideal age is set to zero at the surface
and ages at a rate of 1 yr yr$^{-1}$ in the ocean interior. Constant stoichiometry relates the C, N, P, and DFe content of the biological
variables and their exchanges to inorganic variables ($NO_3$, $PO_4$, DFe, $O_2$, ALK, and DIC). The maximum phytoplankton growth
and microbial remineralization rates are assumed to increase with seawater temperature (Eppley, 1972). The iron cycle that is
incorporated in the biogeochemical model mainly follows Parekh et al. (2005). In addition to dust deposition, which is assumed
as the only DFe source in Parekh et al. (2005), sedimentary and hydrothermal DFe inputs are considered. When the DFe
concentration exceeds the total ligand concentration, a formulation for the DFe scavenging rate of Moore and Braucher (2008)
is applied. To obtain a realistic distribution of the iron-limited region, total ligand concentration, which controls the amount of
the free form of iron, is set to a global constant value of 0.6 µmol m$^{-3}$ instead of the original value of 1 µmol m$^{-3}$ (Fig. 1a).

Dust deposition flux is obtained from the monthly output data of MIROC-ESM in the PI and LGM simulations (Ohgaito et al.,
2018). Dust is assumed to contain a constant fraction of iron (3.5 wt%); 1% of the iron in desert dust is assumed to
instantaneously dissolve at the sea surface. The global DFe flux from dust in the PI is 2.7 Gmol yr$^{-1}$ (Table 1). We used two
sets of LGM dust deposition flux labelled as LGMctl and LGMglac as calculated in a previous study (Ohgaito et al., 2018).
LGMctl is the standard LGM simulation, which has been submitted to Coupled Model Intercomparison Project Phase 5 /
Paleoclimate Modelling Intercomparison Project (CMIP5/PMIP4). LGMglac is identical to LGMctl, except that an additional
glaciogenic dust flux based on Mahowald et al. (2006) is included. In LGMctl, the dust deposition flux is underestimated in
North America, Eurasia, the South Pacific, the SO, and Antarctica compared to the proxy data of ice and sediment cores
(Kohfeld et al., 2013; Albani et al., 2014). Because glaciogenic dust increases dust deposition at high latitudes, the
underestimation is generally improved in LGMglac (see Ohgaito et al., 2018, for more details). The global DFe fluxes from
dust are 8.6 Gmol yr$^{-1}$ and 13.9 Gmol yr$^{-1}$ for LGMctl and LGMglac, respectively.

Present observation generally shows a lower Fe solubility at a higher Fe concentration in aerosols and a higher solubility at a
lower concentration (Fig. S1). A wider range of aerosol Fe solubility (from 0.2% to 48%) has been derived from observations
over the SO, but different types of Fe-containing minerals such as pyrogenic Fe oxides can be considered to achieve high Fe
solubilities (Ito et al., 2019). Thus, an assumed constant iron solubility of 2% in all types of dust could lead to overestimation
of a total DFe flux from different types of Fe-containing aerosols during the LGM (Muglia et al., 2017). However, a much
higher Fe solubility (1–42% of Fe solubility) as derived from observations for the LGM aerosols in Antarctica has suggested
that an assumed constant iron solubility of 1–2% for all types of dust could lead to a DFe flux underestimation during the LGM
(Conway et al., 2015). In LGM_glac3%, an iron solubility of 3% in glaciogenic dust is assumed (Schroth et al., 2009), such
that the global DFe flux is 24.5 Gmol yr$^{-1}$. This value is approximately 10 times larger than that of the PI simulation and is
larger than a recent estimation, suggesting that a quadrupling of the global DFe flux is constrained by a model-proxy
comparison of $\delta^{15}N$ and $\delta^{13}C$ (Muglia et al., 2018). As with the present DFe input from dust, glacial DFe input has large
uncertainties. As an upper estimate of the DFe flux from dust, we set the iron solubility at 10% in glaciogenic dust in
LGM_glac10%.

The DFe input flux from the sediments is estimated based on Moore and Braucher (2008). We assumed that the sedimentary
DFe flux is proportional to the flux of organic carbon reaching the sea floor. To consider the realistic bathymetry of the
continental shelves, the iron flux is weighted by the fraction of bottom area of the ETOPOV2 data that falls within the bounds
of the model grid cell. The global DFe flux from the sediments in the PI is 33.1 Gmol yr$^{-1}$. In the LGM simulations, the DFe
input from sedimentary sources changes according to the flux of organic carbon reaching the sea floor. A decrease in the DFe
input from sedimentary sources because of a sea-level drop is not considered. Muglia et al. (2017) showed this effect causes a
$CO_2$ increase of 15 ppm. The hydrothermal DFe flux is regulated by the ridge spreading rate, as parameterized by a constant
DFe/Helium ratio (Tagliabue et al., 2010). The hydrothermal DFe flux in the PI is ~8.5 Gmol yr$^{-1}$. In the LGM simulations,
the DFe input from hydrothermal sources is the same as that from PI.

The biogeochemical model was initialized from annual mean climatology data based on the World Ocean Atlas 2009
(WOA2009: Garcia et al., 2010a and 2010b) for dissolved $NO_3$, $PO_4$, and $O_2$ and the Global Ocean Data Analysis Project (Key
et al., 2004) for DIC and ALK. The initial DFe concentration is a constant value of 0.6 nM. For the spin-up, the last 50 years
of data in the MIROC PI experiments were cyclically applied to the offline ocean biogeochemical model. The model was spun
up for more than 3000 years with a prescribed atmospheric $CO_2$ concentration of 285 ppm to eliminate model drift in the global
inventory of all tracers. Similar to Yamamoto et al. (2015), all physical and biogeochemical tracers, except salinity and
dissolved iron, have correlation coefficients with observational data greater than 0.85 and normalized standard deviation values
between 0.7 and 1.1.

LGM experiments were run for 3000 years, following 3000 years of spin-up under PI conditions. The atmospheric $CO_2$
concentration was predicted. We increased the salinity, $PO_4$, and $NO_3$ inventory by 3% to account for the reduced ocean
volume because of the sea level drop. All experiments are listed in Table 1. LGM_clim uses LGM boundary conditions.
LGM_dust is based on LGM_clim but uses the dust deposition flux of LGMctl. Similarly, LGM_glac3% and LGM_glac10%
use the dust deposition flux of LGMglac, but with an iron solubility of glaciogenic dust of 3% and 10%, respectively. LGM_all
is similar to LGM_glac3%, but the $NO_3$ and $PO_4$ inventories are increased by 15%. This assumption is based on a recent model
simulation that shows a ~15% increase in nutrient inventory is caused by reduced organic matter burial in shallow sediments
associated with a sea level drop (Wallmann et al., 2016). In our simulations, changes in benthic denitrification were not
considered. Somes et al. (2017) show that a decrease in benthic denitrification because of a sea level drop reduces $NO_3$ loss
and thus leads to a larger $NO_3$ inventory in the LGM ocean. We analyzed the results from the last 100 years of each simulation.

**3 Results and Discussion**
**3.1 Glacial nutrient cycles and export production**
In the LGM_clim, which uses LGM climate boundary conditions, the $NO_3$ redistribution induced by weaker and shallower
AMOC reduces nutrient supply from the deep ocean to the surface (Table 2 and Fig. 2). The $NO_3$ concentration in the euphotic
zone decreases by 12% and the global export production (EP) is reduced by 0.54 Pg C yr$^{-1}$ compared to that of the PI simulation.
Corresponding to the surface $NO_3$ decrease, significant EP decreases are found in the North Atlantic and North Pacific (Fig.
3a and Fig. S2). However, the surface DFe concentration slightly changes. Because these changes in DFe and $NO_3$ decrease
the iron-limited areas by 27% (Fig. 1b), the simulated LGM climate tends to mitigate the impacts of iron fertilization on
biological productivity and the carbon cycle.

To evaluate the impacts of desert and glaciogenic dust on the ocean biogeochemical cycles, we conducted sensitivity studies.
The DFe input from desert dust with a 1% iron solubility was applied in LGM_dust, whereas glaciogenic dust with 3% or 10%
iron solubility was additionally applied in LGM_glac3% or LGM_glac10%, respectively. Iron fertilization from only desert
dust has a limited impact on the EP. Iron fertilization from both desert and glaciogenic dust increases the EP by 0.88 Pg C yr$^-$
$^1$ south of 45°S whereas the EP decreases by 0.86 Pg C yr$^{-1}$ north of 45°S, where most oceans are nitrogen-limited regions
(LGM_glac3% – LGM_clim; Table 2). Enhanced primary production consumes the $NO_3$ of the euphotic zone in the SO and
its anomaly is transported to the Antarctic bottom water (AABW). Subsequently, the surface $NO_3$ reduction in the SO is also
transported to low-latitude regions via surface and intermediate waters (Fig. 2), thus reducing the EP in nitrogen-limited regions
at low latitudes. Remarkable EP reductions occur north of the iron-limited regions of the SO (Fig. 3b). Our results demonstrate
that enhanced biotic carbon export in the SO is partly compensated for by reduced carbon export in low-latitude regions. From
the comparison between the effect of desert dust (LGM_dust – LGM_clim) and that of glaciogenic dust (LGM_glac3% –
LGM_dust), we found that an increase in the EP due to dust-bone iron fertilization in the SO is mainly caused by glaciogenic
dust (Table 2).

For 15% increases in $NO_3$ and $PO_4$ inventory associated with sea level drop (LGM_all), the EP increases globally in the
nitrogen-limited regions, leading to a global EP increase of 0.86 Pg C yr$^{-1}$ (LGM_all – LGM_glac3%; Table 2). Simulated EP
changes from the PI are in good agreement with the paleoproductivity reconstruction (Kohfeld et al., 2013) (Fig. 3c). Among
the common patterns is the north-south dipole pattern in the SO with an EP decrease at higher latitudes and an EP increase at
lower latitudes. The EP decrease at higher latitudes is attributed to sea ice expansion and the associated reduction of surface
shortwave radiation (Oka et al., 2011) whereas iron fertilization increases the EP at lower latitudes. In the model, the EP
changes also have an east-west dipole pattern; a slight EP increase is found in the South Pacific Ocean and significant EP
increases occur in the South Atlantic and Indian oceans. We found that this pattern is attributed to iron fertilization by
glaciogenic dust. Glaciogenic dust derived from Patagonian glaciers is transported to the South Atlantic and Indian oceans by
the southern westerly wind; however, it is unable to reach the South Pacific (Fig. S3). Proxy data show no clear east-west
dipole pattern, suggesting that the model underestimates iron fertilization in the Pacific sector of the Southern Ocean. However,
proxy data in the South Pacific remain sparse and a quantitative comparison of EP changes between the South Atlantic and
South Pacific is limited. Therefore, further proxy data in the South Pacific is required for a comprehensive understanding of
the glacial EP changes and iron fertilization.

**3.2 $CO_2$ reduction and its relationship to efficiency of the biological pump and dust flux**
Climate change reduces the atmospheric $CO_2$ concentration by 26.4 ppm (LGM_clim – PI, Table 2), which is similar to that
of previous simulations (Chikamoto et al, 2012; Menviel et al., 2012; Kobayashi et al., 2015). Circulation changes (i.e. a
weaker and shallower AMOC and AABW expansion) cause DIC to decrease in the upper ocean and increase below 2000 m
depth, such that the vertical DIC gradient between the surface and deep oceans is enhanced (Fig. 4). The efficiency of the
oceanic biological pump is calculated following Ito and Follows (2005). The global mean preformed $PO_4$ is the difference
between the total globally averaged $PO_4$ and global mean remineralized $PO_4$, $P_{pref} = P_{tot} - P_{remi}$. Here, $P_{pref}$ is the preformed
$PO_4$ concentration, $P_{tot}$ is the total $PO_4$ concentration, and $P_{remi}$ is the remineralized $PO_4$ concentration. The remineralized $PO_4$
is given by $P_{remi}$=AOU $\times R_{P:O}$, where $R_{P:O}$ is a constant phosphorous to oxygen ratio, and AOU is apparent oxygen utilization.
A decrease in preformed $PO_4$ and thus an increase in remineralized $PO_4$ indicate an increase in the efficiency of the oceanic
biological pump. Although globally integrated EP decreases, circulation change and deepening of the remineralization profile
due to seawater cooling (Matsumoto, 2007) reduce the preformed nutrient inventory, enhancing the efficiency of the biological
pump (Table 2). The enhanced accumulation of respired carbon associated with the more efficient biological pump and
increased $CO_2$ solubility from the lower SST contribute to a decreased $CO_2$. Notably, the AOU is different from true oxygen
utilization due to the air-sea disequilibrium which is on the order of 20 mmol m$^{-3}$ in deep-water formation regions (Russell and
Dickson, 2003; Duteil et al., 2013). Changes in surface ocean disequilibrium between the PI and LGM simulations might lead
to large errors in the AOU changes (Khatiwala et al., accepted).

Iron fertilization from desert and glaciogenic dust enhances the vertical DIC gradient and causes a $CO_2$ reduction of 1.2 ppm
(LGM_dust – LGM_clim) and 15.6 ppm (LGM_glac3% – LGM_dust), respectively. Our results show that the glacial $CO_2$
reduction due to dust-bone iron fertilization is mainly driven by glaciogenic dust. A simulated total $CO_2$ reduction of 16.8 ppm
induced by iron fertilization is within the range of previous studies using OGCM or Earth system Models of Intermediate
Complexity (EMICs) (8-25 ppm $CO_2$ drawdown (Bopp et al., 2003; Parekh et al., 2006; Tagliabue et al., 2009; Oka et al.,
2011; Menviel et al., 2012; Lambert et al., 2015; Heinze et al., 2016; Muglia et al., 2017). DFe supply from dust also contributes
to the glacial $CO_2$ reduction through enhanced efficiency of the biological pump (Table 2). The simulated atmospheric $CO_2$
concentration is proportionally reduced to the preformed $PO_4$ (Fig. 5a), similar to previous simulations under the present
climate (Ito and Follows, 2005; Marinov et al., 2008). Figure 5b shows the $CO_2$ change in response to the DFe input magnitude.
The iron fertilization efficiency to reduce $CO_2$ decreases with increasing DFe flux. This nonlinear response is driven by a
decrease in the iron-limited areas and the associated weakening of the iron fertilization effect on EP (Fig. 5c). Because the
iron-limited region dramatically decreases in size and the $CO_2$ difference between LGM_glac3% and LGM_glac10% is small,
the $CO_2$ reduction of 20 ppm in LGM_glac10% is near the upper limit (i.e. there are no iron-limited regions and thus no
additional $CO_2$ reduction).

The simulated upper limit of $CO_2$ reduction resulting from iron fertilization is not a robust result because present iron models
have large uncertainty. While Parekh et al. (2008) show an upper limit of 10 ppm, other simulations show $CO_2$ decrease by
greater than 20 ppm (Oka et al., 2011; Muglia et al., 2017). To obtain a better understanding of the impact of iron fertilization
on glacial $CO_2$ decrease, the variability of the upper limit among iron models should be investigated in a future study.

Increases in nutrient inventory from lower sea levels drive an additional $CO_2$ drawdown by 16 ppm (LGM_all – LGM_glac3%).
We found that changes in the biological pump induced by iron fertilization and an increase in nutrient inventory contribute to
a glacial $CO_2$ decrease of more than 30 ppm. The resultant total $CO_2$ reduction is ~60 ppm, which our model does not reproduce
as the full variation in the glacial-interglacial $CO_2$ change. Note that changes in the sedimentation process (i.e. carbonate
compensation and burial-nutrient feedback) are not considered in our simulation. The simulated increase in the bottom water
DIC (Fig. 4) would enhance calcium carbonate dissolution in the sediments and thereby increase ocean alkalinity, leading to
a further $CO_2$ decrease (Bouttes et al., 2011; Brovkin et al., 2012; Kobayashi et al., 2018). The long-term balance between the
burial of organic material and nutrient input through weathering is also potentially important for the atmospheric $CO_2$ response
(Roth et al., 2014; Wallmann et al., 2016). For example, Tschumi et al. (2011) show that the nutrient-burial feedback
significantly amplifies the effect of an increase in the $PO_4$ inventory on the glacial $CO_2$ decrease. Menviel et al. (2012)
quantified the implication of ocean-sediment-lithosphere coupling for factorial experiments with an altered iron fertilization
and altered $PO_4$ inventory from transient glacial-interglacial simulations. Considering that EP increases due to iron fertilization
and the nutrient increase are smaller in our simulations than that in previous studies (Tschumi et al., 2011; Menviel et al.,
2012), the effect of burial-nutrient feedback on the glacial $CO_2$ reduction may be smaller than previously estimated. As
described in the next section, to assess the simulated accumulation of respired carbon, we compared the simulated oxygen
changes to qualitative and quantitative proxy records.

**3.3 Model-proxy comparison of glacial oxygen changes**
Compared to the compilation of qualitative and quantitative proxy records of oxygen change from the Holocene to Last Glacial
Maximum, LGM_clim shows an increase in oxygen for the entire SO and underestimates deoxygenation in the deep Pacific
and Indian oceans, in contrast to the proxy records (Fig. 6a). However, LGM_all successfully reproduces large-scale spatial
patterns of oxygen change, including for the SO (Fig. 6b). Moreover, the simulated changes in oxygen concentration agree
well with quantitative reconstructions: a 45-65 mmol $m^{-3}$ decrease in the deep North Atlantic (Hoogakker et al., 2015), an ~30-
80 mmol $m^{-3}$ decrease in the eastern equatorial Pacific (Hoogakker et al., 2018; Umling and Thunell, 2018), and a >80 mmol
$m^{-3}$ in the upper SO of the Pacific sector (Lu et al., 2016). Our results clearly show the importance of iron fertilization and an
increase in nutrient inventory in global deep deoxygenation. These model-proxy agreements of oxygen change support the
simulated $CO_2$ decrease of 30 ppm by the biological pump. However, the reconstructed $O_2$ decrease of ~175 mmol $m^{-3}$ in the
deep SO (Gottschalk et al., 2016) is much greater than the simulated decrease of ~30 mmol $m^{-3}$ from LGM_all; thus, the
respired carbon accumulation in the deep SO is underestimated in our model. This may be one of the reasons why the glacial-
interglacial $CO_2$ change of ~100 ppm cannot be reproduced in our simulations.

To clarify the mechanism of $O_2$ change from LGM_all – PI, we decomposed the $O_2$ change into changes in saturation ($O_{2sat}$)
and apparent oxygen utilization (AOU), where $\Delta O_2 = \Delta O_{2sat} - \Delta AOU$. $O_{2sat}$ is computed from simulated seawater temperature
and salinity and AOU by subtracting the $O_2$ concentration from $O_{2sat}$. Ocean cooling increases $O_{2sat}$ globally, increasing the
global mean value by 25.5 mmol m$^{-3}$ (Fig. 7a). As with the $O_2$ change, $\Delta AOU$ shows a contrast between the upper and deep
oceans (Fig. 7b). At a depth of 0-800 m, the AOU decreases by 5.2 mmol m$^{-3}$ north of 45°S, which results from the decrease
in biological oxygen consumption associated with EP reduction and increased ventilation (Fig. 7f). Therefore, the combined
effects of an $O_{2sat}$ increase and AOU decrease contribute to an overall $O_2$ increase in the upper ocean. In the deep ocean (>2
km depth), the sum of AOU increases by 72.8 mmol m$^{-3}$ (LGM_all in Table 2), overcoming the $O_{2sat}$ increase, resulting in
deep $O_2$ depletion. The relationship between changes in the $O_2$ concentration, $O_{2sat}$, and AOU are consistent with that of a
previous simulation (Bopp et al., 2017).

The $\Delta AOU$ is also decomposed into effects of climate change (LGM_clim – PI), iron fertilization (LGM_glac3% – LGM_clim)
and an increase in nutrient inventory (LGM_all – LGM_glac3%). The effects of climate change, circulation change, restricted
air-sea gas exchange from sea-ice expansion, and deepening of remineralization due to seawater cooling leads to the AOU
increasing by 37.3 mmol m$^{-3}$ in the deep ocean (Table 2). In the deep North Atlantic, the simulated water mass age is older in
the LGM than in the PI by up to 500 years, suggesting reduced ventilation (Fig. 7f). Therefore, significant AOU increases
occur (Fig. 7c). Meanwhile, in the SO and deep Pacific Ocean, an increase in ventilation tends to decrease the AOU and thus
partly compensates for the increase in the AOU. Regarding the effects of iron fertilization and nutrient inventory, the EP
changes associated with iron fertilization and an increase in nutrient inventory enhance biological oxygen consumption and
thus increase the AOU by 21.4 and 14.1 mmol m$^{-3}$ in the deep ocean, respectively (Table 2 and Fig. 7d, e). In particular,
glaciogenic dust causes an increase in the AOU of 19.8 mmol m$^{-3}$. Our results demonstrate that in addition to climate change,
enhanced biological oxygen consumption associated with iron fertilization and increased nutrient inventory are crucial drivers
of glacial deoxygenation in the deep ocean. While some previous modelling studies show deep ocean oxygenation during the
LGM (Buchanan et al., 2016; Galbraith and Lavergne, 2018), this study and others reproduce deep ocean deoxygenation
(Galbraith and Jaccard, 2015; Schmittner and Somes, 2016; Bopp et al., 2017; Somes et al., 2017). The conflicting oxygen
change between the previous simulations can be attributed to different treatments of enhanced biological oxygen consumption
because iron fertilization and increased nutrient inventory are not considered in these simulations that fail to reproduce deep
deoxygenation (Buchanan et al., 2016; Galbraith and Lavergne, 2018).

Glacial oxygen change expands the volume of hypoxic waters (defined here as $[O_2]$ <80 mmol m$^{-3}$) below 1000 m depth, such
that the simulated global volume increases from the present value of 120 Mkm$^3$ to 237 Mkm$^3$ in LGM_all. Significant
expansion occurs in the deep Pacific and Indian oceans (Fig. 8), with hypoxic waters also appearing in the upper SO in the
Pacific sector, consistent with proxy records (Hoogakker et al., 2018; Lu et al., 2016). Because hypoxic conditions are lethal
for more than one-half of marine benthic animals (Vaquer-Sunyer and Duarte, 2008), expansion of hypoxic water in the deep
ocean can have an adverse impact on benthic fauna. Determining the biotic responses to glacial expansion of hypoxic water
would be helpful for understanding the biotic response to future deoxygenation associated with global warming.

Finally, we discuss the underestimation of deoxygenation in the deep SO in LGM_all. Because simulated changes in the
biological pump and sea-ice distributions are consistent with reconstructions (Obase et al., 2017), we then addressed circulation
changes. The simulated water mass age of the deep SO is younger during the LGM than during the PI by ~200 years (Fig. 7f),
indicating an increase in ventilation. However, $\Delta^{14}C$ records show an increase in water mass age of more than 1000 years, and
thus increased stratification (Skinner et al., 2010; Burke and Robinson, 2012). Enhanced mixing of surface waters with deep
waters supplies oxygen-rich surface waters to the deep ocean and simultaneously releases carbon accumulated in the deep
water to the atmosphere. Therefore, we attribute the underestimation of deoxygenation and carbon accumulation in the deep
SO to overestimated ventilation. Our results suggest that a stratified SO is required for reproducing glacial $CO_2$ drawdown and
oxygen decline in the deep SO, consistent with recent paleo-proxy data and models (Fischer et al., 2010; Sigman et al., 2010;
Kobayashi et al., 2015; Menviel et al., 2017). Menviel et al (2017) showed that weaker and shallower North Atlantic Deep
Water and weaker AABW could be necessary to reproduce the LGM oceanic $\delta^{13}C$ and radiocarbon distribution.

**4 Conclusion and remarks**
We quantified the impacts on glacial deoxygenation and $CO_2$ decreases caused by glaciogenic dust with higher iron solubility
and increase in nutrient inventory associated with a sea-level drop using the coupled atmosphere–ocean general circulation
model, aerosol model, and ocean biogeochemical model. As a result, we successfully reproduced the magnitude and large-
scale pattern of the observed oxygen change between the present and LGM. Our results show that iron fertilization from
glaciogenic dust and an increase in nutrient inventory could explain a glacial $CO_2$ decline of more than 30 ppm and
approximately one-half of deep ocean deoxygenation. These results also demonstrate the usefulness of the quantitative model-
proxy comparison of oxygen change in understanding glacial-interglacial $CO_2$ change. However, large uncertainty remains
because of the limited number of proxy data of quantitative oxygen change. Thus, we anticipate our findings will encourage
studies to obtain further qualitative and quantitative reconstructions from throughout the global deep ocean. A comparison
between the models and other proxy data (e.g. $\delta^{13}C$, (Schmittner and Somes, 2016)) is also required to obtain a more robust
and comprehensive understanding of the glacial carbon cycle.

The changes in nutrient inventory during the LGM have large uncertainties. Previous studies estimate that the oceanic $PO_4$ and
$NO_3$ inventories could have been 15–40% (Tamburini and Föllmi, 2009; Wallmann et al., 2016) and 10-100% (Deutsch et al.,
2004; Eugster et al., 2013; Somes et al., 2017) greater during glacial compared to interglacial periods, respectively. Moreover,
Somes et al. (2017) shows that sedimentary $\delta^{15}N$ records provide no constrain on this effect. Future simulations should test the
biogeochemical sensitivity to nutrient inventory changes.

We focused on the impacts of DFe flux changes from the dust on glacial $CO_2$ drawdown and deoxygenation in this study.
However, changes in the sedimentary and hydrothermal DFe flux and ligand concentration that are not considered in this study
could also be important. A glacial sea-level drop decreases the sedimentary DFe flux due to the continental shelf reduction.
However, the hydrothermal DFe flux is increased by the lower sea level and bottom pressure (Middleton et al., 2016). Muglia
et al. (2017) show that the changes in sedimentary and hydrothermal DFe flux associated with a sea-level drop increase $CO_2$
by 15 ppm and decrease $CO_2$ by 6 ppm, respectively. Although sedimentary DFe flux is proportional to the organic carbon
flux reaching the seafloor in our model, a parametrization with the Dfe flux as a function of organic carbon flux and bottom
oxygen concentrations is proposed in Dale et al. (2015). Glacial deep-water deoxygenation would increase sedimentary DFe
flux, leading to a further $CO_2$ decrease via the biological pump. Ligand concentrations strongly control DFe concentrations
(Gledhill and Buck, 2012). Because the ligand concentration is affected by numerous factors (Völker and Tagliabue, 2015),
changes in ligand concentration from the PI to LGM have large uncertainty. Thus, we quantified the effect of DFe flux changes
under a constant ligand concentration in the PI and LGM simulations. Changes in the sedimentary and hydrothermal DFe flux
and ligand concentration should be the subject of future research.

Our model-proxy comparison shows the importance of the combination of a more sluggish SO circulation and enhanced
biological transport of organic matter in the increased accumulation of respired carbon and deoxygenation in the deep SO.
However, present climate models cannot reproduce the stratified SO. A possible reason is that they are too coarse to capture
the process of dense water formation on the Antarctic shelf and tend to underestimate the strength of stratification in the SO
(Heuzé et al., 2013). The brine rejection process and/or change in the vertical diffusion coefficient could be necessary to
reproduce the stratified SO (Kobayashi et al., 2015; Bouttes et al., 2011). Similar to glacial oxygen changes, changes in ocean
circulation in the SO are crucial in projecting future oxygen changes associated with global warming (Yamamoto et al., 2015).
Therefore, an understanding of glacial oxygen changes will aid in better understanding and predicting future oxygen changes.

**Data availability.** Data are freely available from the corresponding author ([akitomo@jamstec.go.jp](mailto:akitomo@jamstec.go.jp)) upon request.

**Author contributions.** AY and AA-O designed the research. AY conducted, and analysed experiments and prepared the paper.
AA-O provided the results of MIROC. RO provided the results of MIROC-ESM. All authors discussed the results and gave
their inputs on the manuscript.

**Acknowledgements**
We thank the Editor Laurie Menviel as well as Fortunat Joos and Andreas Schmittner for their helpful comments. This work
was supported by the Integrated Research Program for Advancing Climate Models (TOUGOU programme) from the
Ministry of Education, Culture, Sports, Science and Technology (MEXT), Japan, and JSPS KAKENHI grant number
17H06323. The simulations with the offline biogeochemical model were performed using the Fujitsu PRIMEHPC FX10
system in the Information Technology Center, University of Tokyo.

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

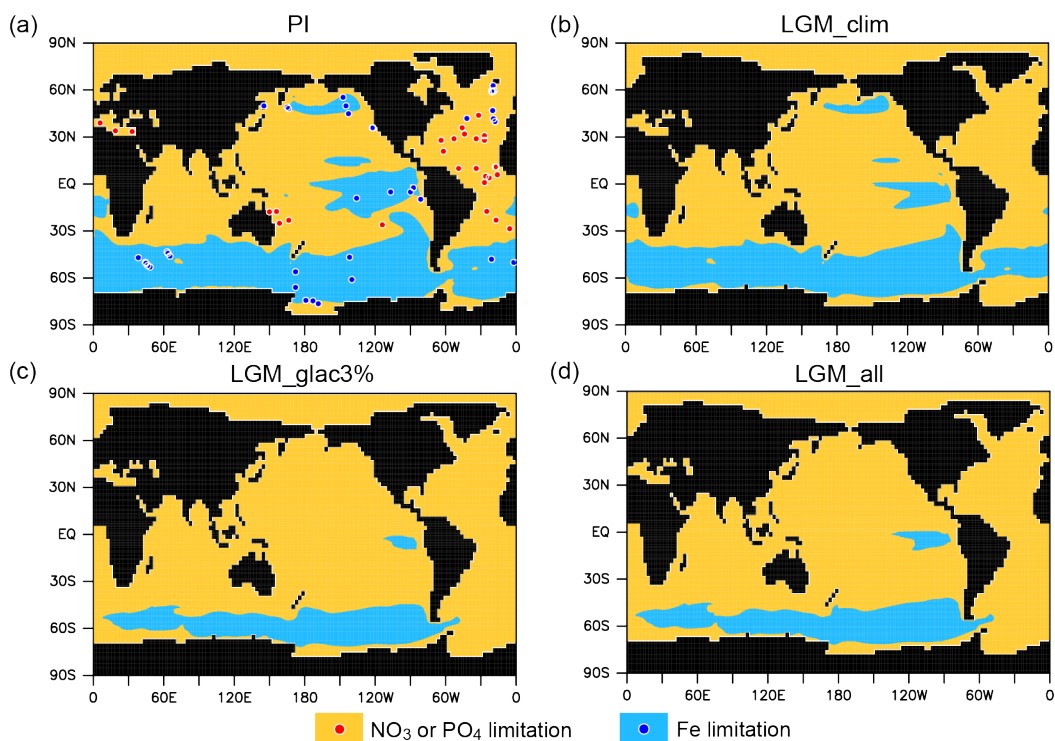


**Figure 1.** Primary limiting nutrient for phytoplankton for the (a) PI, (b) LGM_clim, (c)LGM_glac3%, and (d) LGM_all.
Shade indicates $NO_3$ or $PO_4$ limitation (orange) and Fe limitation (blue). Circles represent observed limiting nutrients from
nutrient addition experiments (Moore et al., 2013).


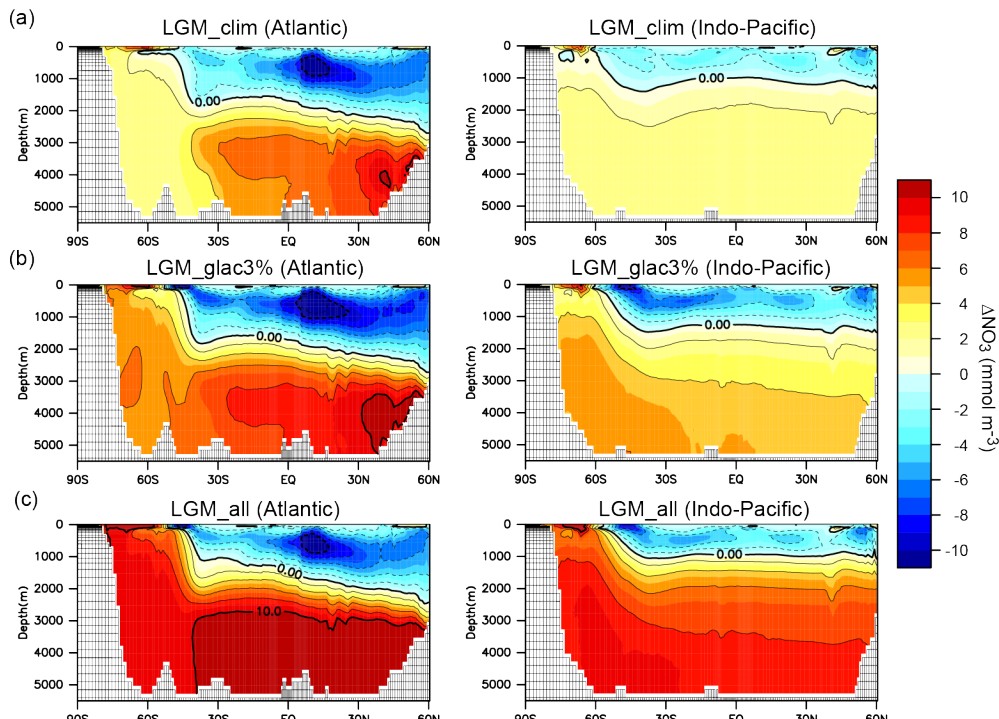

**Figure 2.** NO₃ change resulting from changes in the climate and biological pump in LGM simulations. Zonal mean changes in NO₃ from the PI to (a) LGM_clim, (b) LGM_glac3%, and (c) LGM_all. The left and right panels show the Atlantic and Indo-Pacific oceans, respectively.


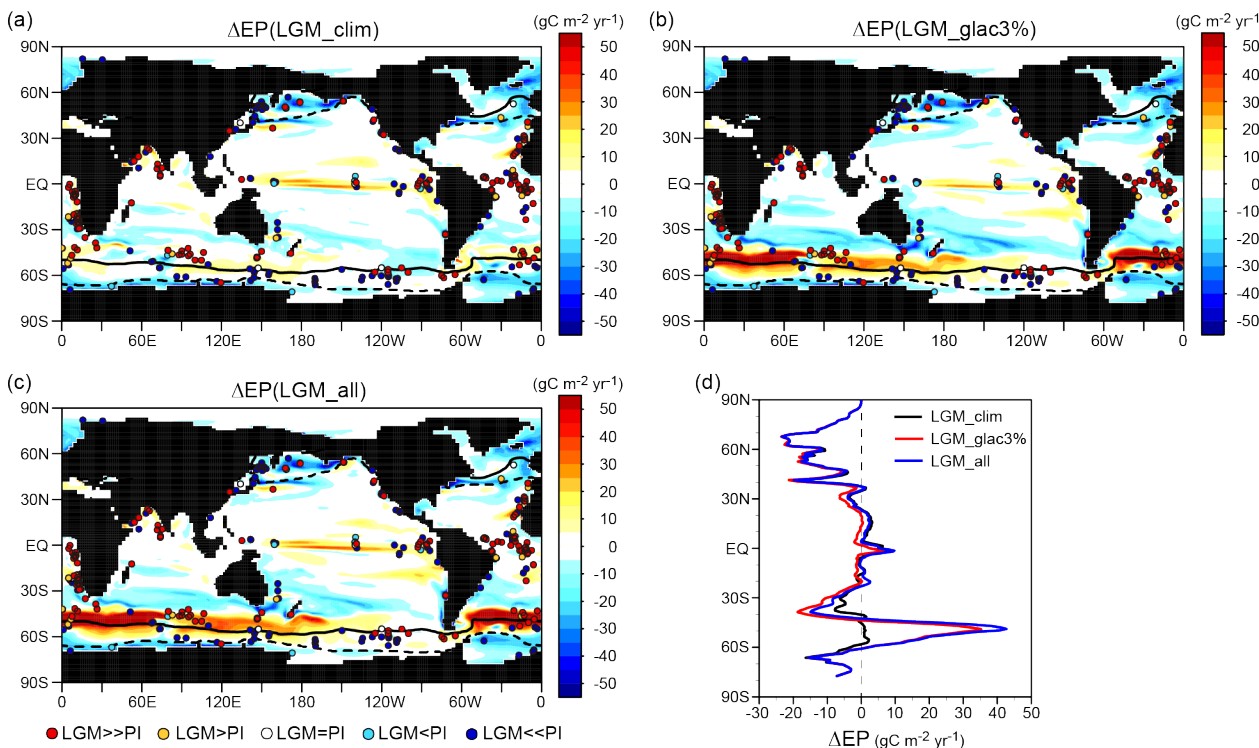


**Figure 3.** Model-proxy comparison of EP change from the PI to LGM. The EP difference from the PI for (a) LGM_clim, (b)


LGM_glac3%, and (c) LGM_all. Circles show proxy data (Kohfeld et al., 2013). Solid (dotted) lines refer to the glacial sea


ice fraction of 0.1 during August (February). (d) Zonal mean changes in the surface EP from the PI for LGM_clim (black),


LGM_glac3% (red), and LGM_all (blue).



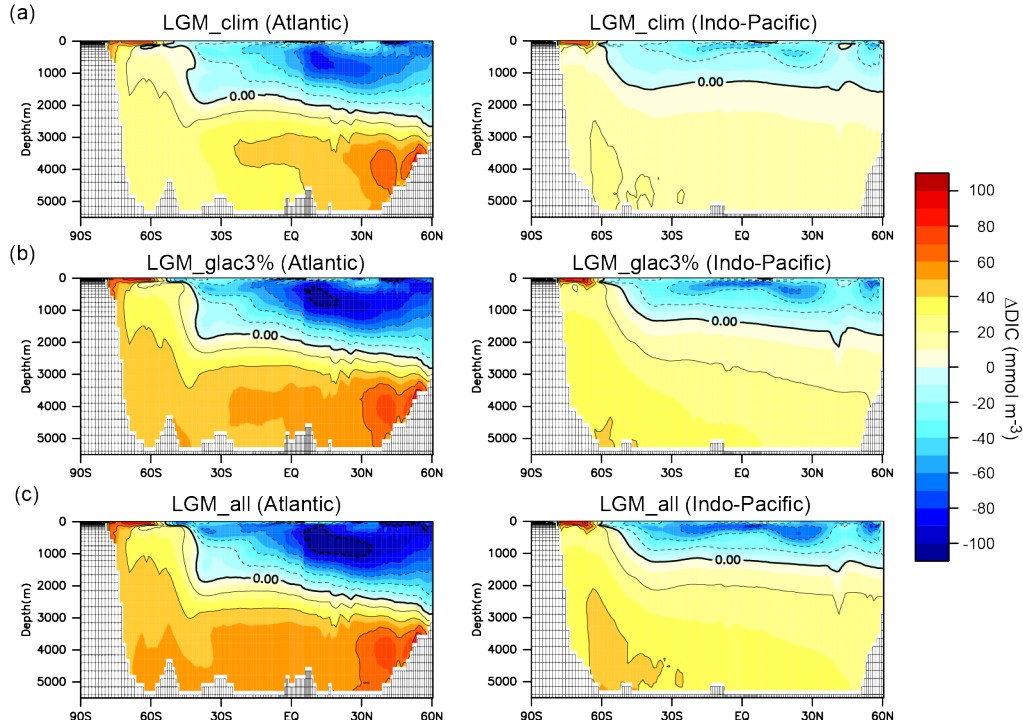


**Figure 4.** DIC change resulting from changes in the climate and biological pump in LGM simulations. Zonal mean changes
in DIC from PI to (a) LGM_clim, (b) LGM_glac3%, and (c) LGM_all. The left and right panels show the Atlantic and Indo-
Pacific oceans, respectively.

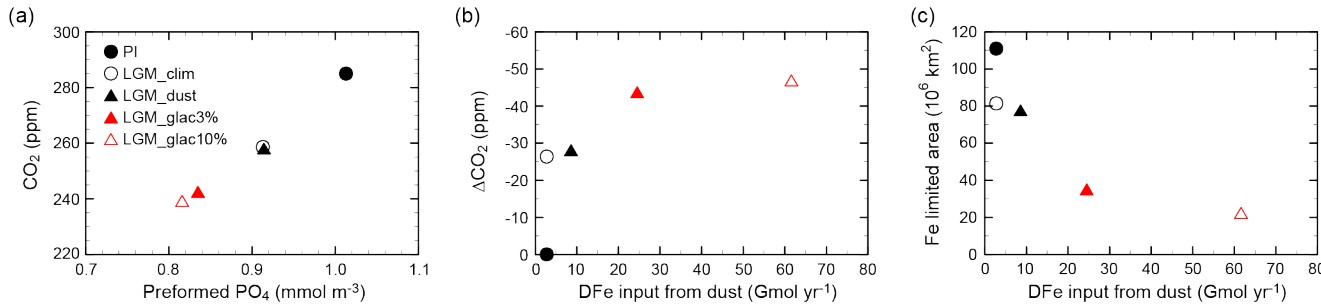


**Figure 5.** $CO_2$ change and its relationship to efficiency of the biological pump and iron cycle. (a) Atmospheric $CO_2$ as a
function of globally averaged preformed $PO_4$. (b) Changes in $CO_2$ from the PI as a function of DFe input from dust. (c) Fe-
limited area as a function of DFe input from dust. Shown are the PI (black filled circle), LGM_clim (black open circle),
LGM_dust (black filled triangle), LGM_glac3% (red filled triangle), and LGM_glac10% (red open triangle).


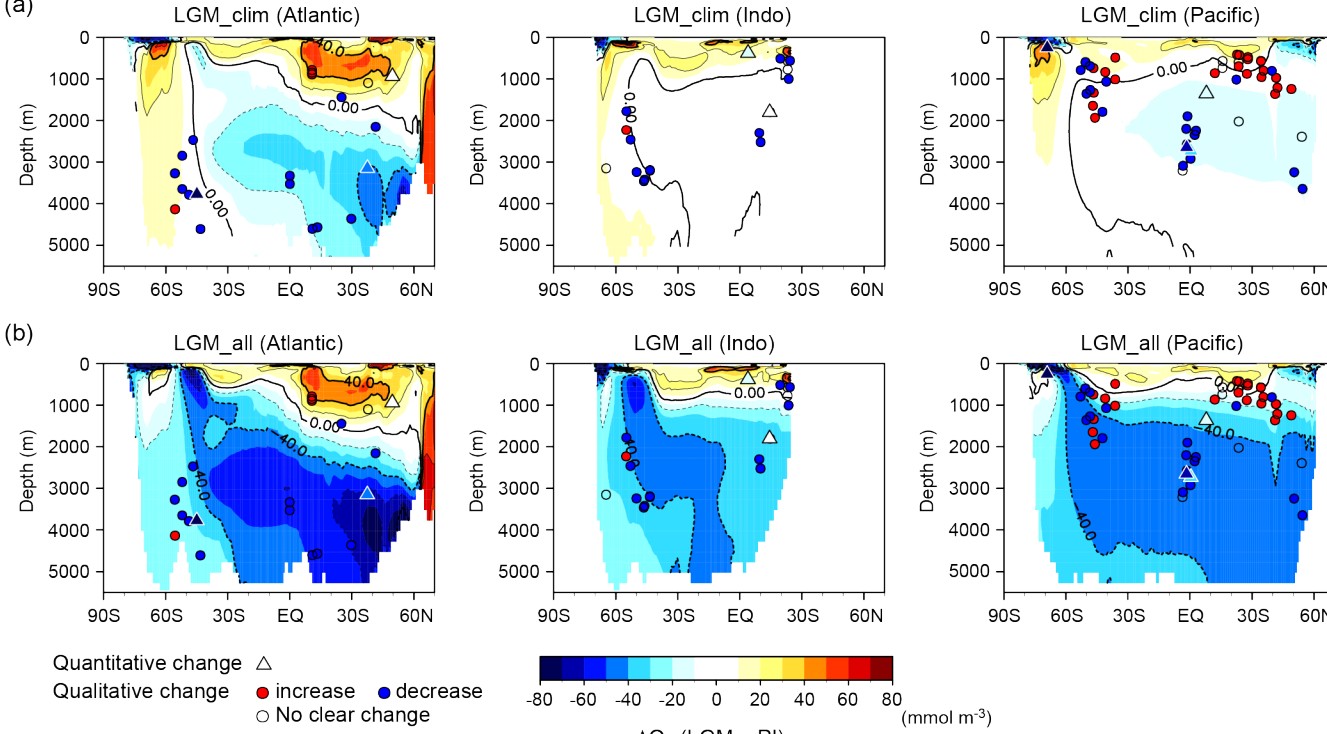

**Figure 6.** Model-proxy comparison of changes in dissolved oxygen concentration from the PI to LGM. Zonal mean changes in $O_2$ from the PI to (a) LGM_clim and (b) LGM_all for the Atlantic (left), Indian (middle), and Pacific (right) oceans; the contour interval is 20 mmol m$^{-3}$. Circles show proxy records of qualitative $O_2$ change from multi-proxy data compilation from Jaccard and Galbraith (2012) (except $\delta^{15}$N data), Jaccard et al. (2016), and Durand et al. (2018). Red (blue) circles indicate $O_2$ increase (decrease) from the Holocene to LGM. Triangles show proxy records of quantitative $O_2$ change from (Schmiedl and Mackensen, 2006; Hoogakker et al., 2015, 2018; Gottschalk et al., 2016; Lu et al., 2016; Bunzel et al., 2017; Umling and Thunell, 2018) (triangles shaded using the same colour scale).



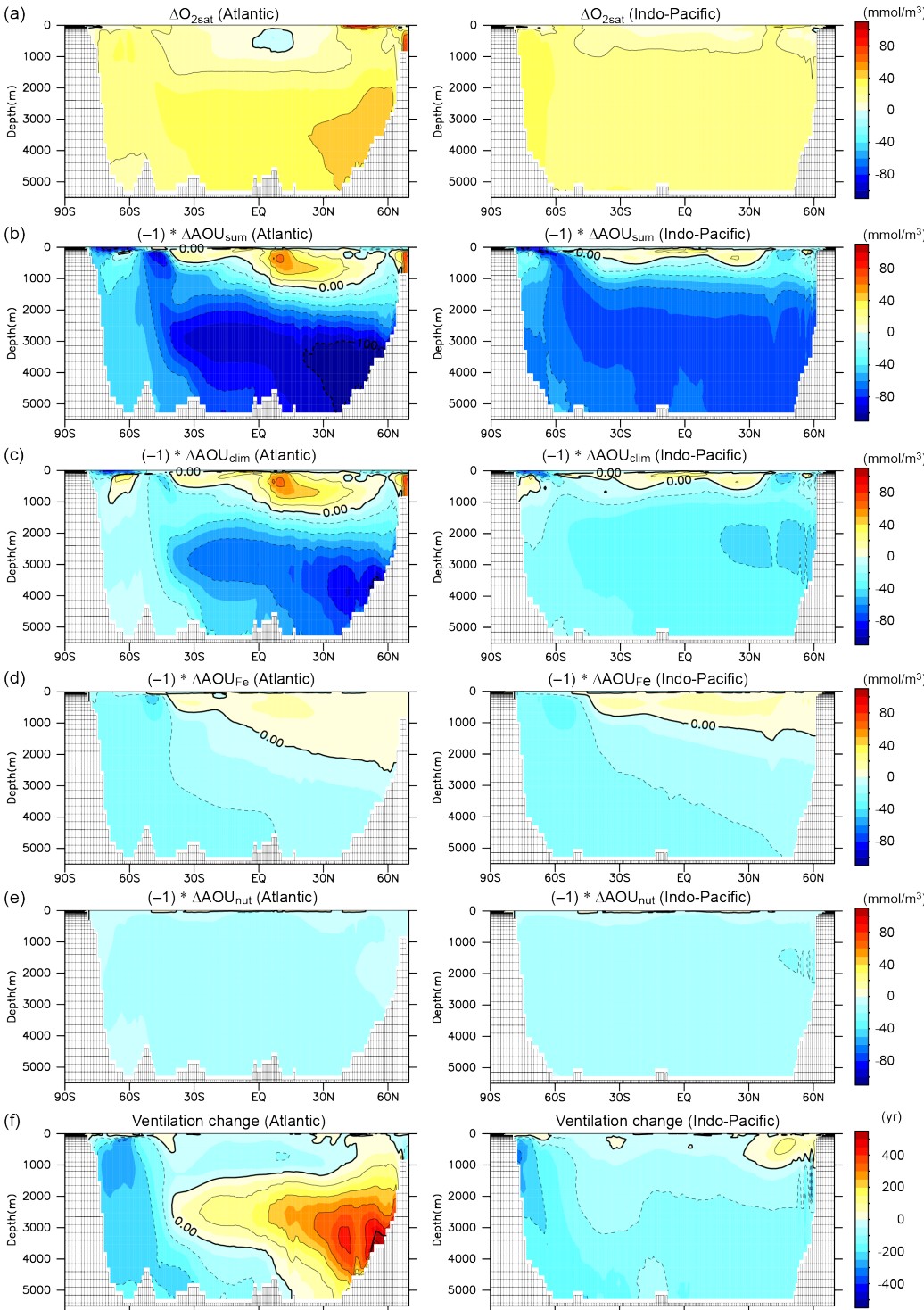


**Figure 7.** Contributions of individual mechanisms to oxygen change and ventilation change. Zonal mean changes of (a) $O_{2sat}$,
(b) $AOU_{sum}$, (c) $AOU_{clim}$, (d) $AOU_{Fe}$, (e) $AOU_{nut}$, and (f) ventilation age from the PI to LGM. Left and right panels show the
Atlantic and Indo-Pacific oceans: the contour intervals are 20 mmol m$^{-3}$ for (a)–(e) and 100 years for (f). We decomposed the
total AOU change ($\Delta AOU_{sum} = AOU_{(LGM\_all)} - AOU_{(PI)}$) into the effects of climate change ($\Delta AOU_{clim} = AOU_{(LGM\_clim)} -$
$AOU_{(PI)}$), iron fertilization ($\Delta AOU_{Fe} = AOU_{(LGM\_glac3\%)} - AOU_{(LGM\_clim)}$), and nutrient inventory increase ($\Delta AOUnut =$
$AOU_{(LGM\_all)} - AOU_{(LGM\_glac3\%)}$).

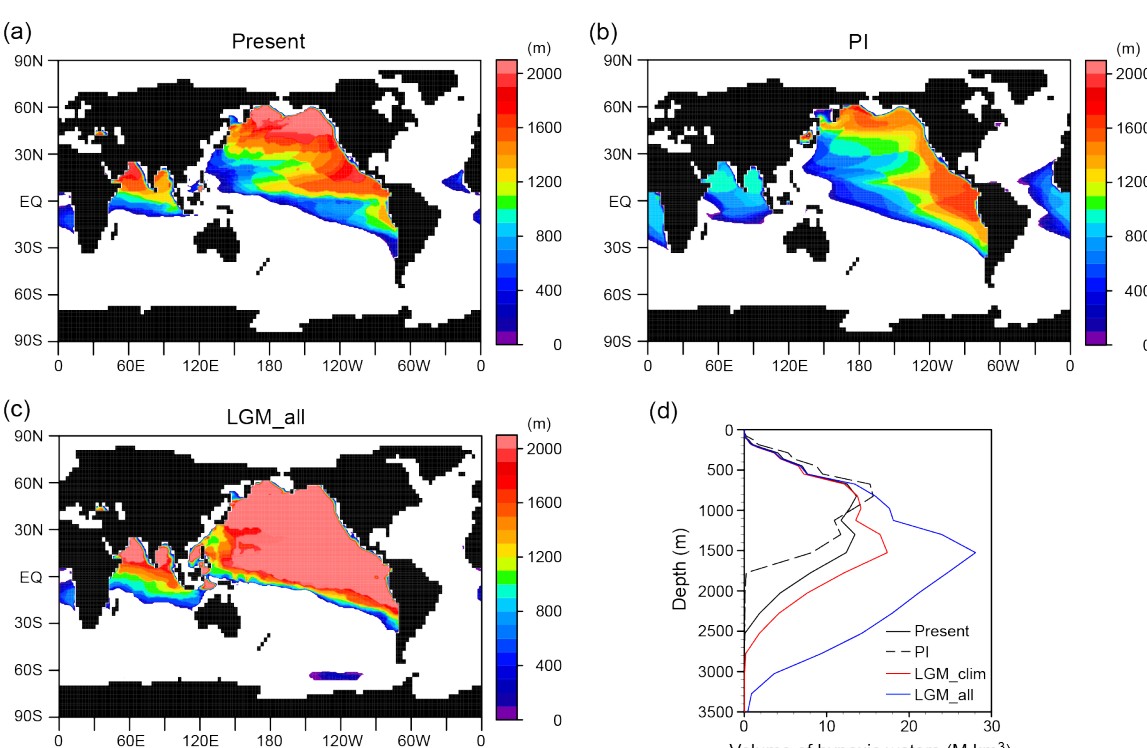


**Figure 8.** Hypoxic waters expansion. Horizontal distribution of thickness of the hypoxic waters ([$O_2$] <80 mmol m$^{-3}$) for the
(a) present, (b) PI, and (c) LGM_all. (d) Vertical distribution of hypoxic waters for the present (black solid), PI (black dashed),
LGM_clim (red), and LGM_all (blue). Because current coarse resolution models have difficulties reproducing low oxygen
concentration for the present day (Bopp et al., 2013), observed values from WOA2009 (Garcia et al., 2010a) were used for the
present. For the LGM simulations, we combined the observed values with the modelled changes.

| Experiments | Climate | Dust deposition | Fe solubility in glaciogenic dust | Dust DFe (Gmol yr$^{-1}$) | Global PO$_4$ (mmol m$^{-3}$) |
|---|---|---|---|---|---|
| PI | PI | PI | - | 2.7 | 2.13 |
| LGM_clim | LGM | PI | - | 2.7 | 2.2 (+3%) |
| LGM_dust | LGM | LGMctl | - | 8.6 | 2.2 (+3%) |
| LGM_glac3% | LGM | LGMglac | 3% | 24.5 | 2.2 (+3%) |
| LGM_glac10% | LGM | LGMglac | 10% | 61.6 | 2.2 (+3%) |
| LGM_all | LGM | LGMglac | 3% | 24.5 | 2.45 (+15%) |


Table 1. Description of the model experiments.

| Experiments | Surface NO$_3$ (mmol m$^{-3}$) | Surface DFe (μmol m$^{-3}$) | Fe limited area (10$^6$ km$^2$) | Global ΔEP (Pg C yr$^{-1}$) | ΔEP (>45˚S) (Pg C yr$^{-1}$) | ΔEP (<45˚S) (Pg C yr$^{-1}$) | Preformed PO$_4$ (mmol m$^{-3}$) | ΔCO$_2$ (ppm) | ΔO$_{2\,deep}$ (mmo m$^{-3}$) | ΔAOU$_{deep}$ (mmol m$^{-3}$) |
|---|---|---|---|---|---|---|---|---|---|---|
| PI | 7.7 | 0.38 | 111 | (8.54) | (6.19) | (2.35) | 1.013 | (285) | (156) | (182.5) |
| LGM_clim | 6.8 | 0.39 | 81 | -0.54 | -0.45 | -0.09 | 0.913 | -26.4 | -7 | 37.3 |
| LGM_dust | 6.9 | 0.42 | 80 | -0.54 | -0.49 | -0.05 | 0.914 | -27.6 | -8 | 38.9 |
| LGM_glac3% | 5.8 | 0.5 | 35 | -0.54 | -1.31 | +0.77 | 0.835 | -43.2 | -28 | 58.7 |
| LGM_glac10% | 5.5 | 0.54 | 23 | -0.54 | -1.46 | +0.92 | 0.816 | -46.4 | -33 | 63.6 |
| LGM_all | 6.5 | 0.48 | 39 | +0.32 | -0.63 | +0.95 | 1.002 | -59.2 | -42 | 72.8 |


Table 2. Results of the model experiments. Simulated global average of surface NO$_3$, DFe, and Fe-limited area and changes in
EP at 100 m, atmospheric CO$_2$, and globally averaged preformed PO$_4$, O$_2$ and AOU below 2000 m depth from the PI. Values
in brackets are the PI results.