# Peer review of "Glacial CO2 decrease and deep-water deoxygenation by iron fertilization from glaciogenic dust"

_Climate of the Past, 2019_

## Referee Comment (RC1) · Fortunat Joos (Referee) · 3 Apr 2019

Yamamoto and colleagues present an interesting analysis of glacial change in atmospheric CO2 and marine oxygen. The authors investigate, using a range of factorial analyses, the impacts of glaciogenic iron input and an increased nutrient inventory in the glacial ocean. They apply an offline biogeochemical model for Last Glacial Maximum (LGM) and preindustrial (PI) conditions. They simulate an upper limit for the CO2 decrease due to iron fertilization of 20 ppm and a similar decrease due to an increase in whole ocean nutrient inventory. They present a novel model-proxy comparison for PI-LGM changes in O2. The results suggest a role of iron fertilization and changes in

nutrient inventory for low glacial CO2 and for the reconstructed oxygen changes.

The manuscript is concise and well written. Figures and tables are illustrative and support the conclusions.

I recommend publication of the manuscript after minor revision.

1) I find it interesting that the upper limit for iron fertilization is 20 ppm (p10, l215). I would appreciate if this finding is lifted to the abstract.

2) Figure 8 shows results from WOA2009 and simulated anomalies. Results for the model for the modern ocean should be displayed as well. This would permit the reader to assess the quality of the simulated O2 field.

3) There are some language problems, e.g. missing articles, and the manuscript would benefit from proof-reading by a native speaker.

4) There is no discussion on the role of the burial-nutrient feedback and how burial-nutrient feedback may affect the results of this study. On page 10, l221, it is mentioned that CaCO3 compensation is not included. However, this study does also not consider how changes in iron fertilization affect the balance between weathering and burial of organic matter. This also applies to some extent to the experiment with the increase in whole ocean nutrient inventory.

Several studies point to the potentially important role of the ocean/sediment/lithosphere fluxes of organic matter and how the associated burial-nutrient feedback modifies the magnitude and time scales of the response in CO2 and other tracers to changes in the marine biological cycles (Wallmann et al., 2016;Roth et al., 2014;Jeltsch-Thömmes et al., 2018). (Tschumi et al., 2011), for example, quantify the implication of ocean-sediment-lithosphere coupling for an experiment where the ocean P inventory is increased. (Menviel et al., 2012) present results from factorial experiments with altered iron fertilization/dust input and altered P inventory plus variation in other drivers from transient glacial-interglacial simulations. I suggest that this caveat is addressed on

page 10 and perhaps also in the discussion section.

Minor and technical comments:

1) P1, line 11, p3, l46: ".. due to sea surface cooling" What matters is in my opinion the cooling of the whole ocean, including the ocean interior. Please modify the wording

2) P1, l16-18: This sentence is not so clear. The circulation changes itself likely induce a change in the efficiency of the biological pump (Volk and Hoffert, 1985) as may also be seen when looking at preformed/remineralized nutrients or AOU. I think it should rather read "whereas the other half is driven by iron fertilization and an increase in whole ocean P inventory" or similar.

3) P5, l90: Is convection included in the offline model and how is this done?

4) P9, l192, You may also refer to (Menviel et al., 2012)

5) P8, l182: missing word: "shortwave radiation"

6) P10, l207: you may include here EMICs results (e.g. (Muglia et al., 2017;Parekh et al., 2008;Menviel et al., 2012;Heinze et al., 2016).

References:

Heinze, C., Hoogakker, B. A. A., and Winguth, A.: Ocean carbon cycling during the past 130 000 years – a pilot study on inverse palaeoclimate record modelling, Clim. Past, 12, 1949-1978, 10.5194/cp-12-1949-2016, 2016.

Jeltsch-Thömmes, A., Battaglia, G., Cartapanis, O., Jaccard, S. L., and Joos, F.: A large increase in the carbon inventory of the land biosphere since the Last Glacial Maximum: constraints from multi-proxy data, Clim. Past Discuss., 2018, 1-48, 10.5194/cp-2018-167, 2018.

Menviel, L., Joos, F., and Ritz, S. P.: Simulating atmospheric CO2, 13C and the marine carbon cycle during the Last Glacial–Interglacial cycle: possible role for a deepening

of the mean remineralization depth and an increase in the oceanic nutrient inventory, Quat. Sci. Rev., 56, 46-68, 10.1016/j.quascirev.2012.09.012, 2012.

Muglia, J., Somes, C. J., Nickelsen, L., and Schmittner, A.: Combined Effects of Atmospheric and Seafloor Iron Fluxes to the Glacial Ocean, Paleoceanography, 32, 1204-1218, 10.1002/2016pa003077, 2017.

Parekh, P., Joos, F., and Müller, S. A.: A modeling assessment of the interplay between aeolian iron fluxes and iron-binding ligands in controlling carbon dioxide fluctuations during Antarctic warm events, Paleoceanography, 23, PA4202, 4201-4214, 10.1029/2007PA001531, 2008.

Roth, R., Ritz, S. P., and Joos, F.: Burial-nutrient feedbacks amplify the sensitivity of atmospheric carbon dioxide to changes in organic matter remineralisation, Earth Syst. Dynam., 5, 321-343, 10.5194/esd-5-321-2014, 2014.

Tschumi, T., Joos, F., Gehlen, M., and Heinze, C.: Deep ocean ventilation, carbon isotopes, marine sedimentation and the deglacial CO2 rise, Clim. Past, 7, 771-800, 10.5194/cp-7-771-2011, 2011.

Volk, T., and Hoffert, M. I.: Ocean Carbon Pumps: Analysis of Relative Strengths and Efficiencies in Ocean-Driven Atmospheric CO2 Changes, in: The Carbon Cycle and Atmospheric CO2: Natural Variations Archean to Present, American Geophysical Union, 99-110, 1985.

Wallmann, K., Schneider, B., and Sarnthein, M.: Effects of eustatic sea-level change, ocean dynamics, and nutrient utilization on atmospheric pCO2 and seawater composition over the last 130 000 years: a model study, Clim. Past, 12, 339-375, 10.5194/cp-12-339-2016, 2016.

---

## Referee Comment (RC2) · Andreas Schmittner (Referee) · 4 Apr 2019

Yamamoto and co-workers present a nice modeling study of glacial ocean oxygen and carbon changes. The manuscript is well written (except for a few typos) and nicely illustrated. I think the main new finding is that glaciological iron sources from Patagonia are particularly important for lowering atmospheric CO2. Although similar suggestions have been made previously with simpler models (e.g. Brovkin et al., 2007) this study is the first to my knowledge that cleanly separates glaciological from other (desert) dust sources.

However, I have a few concerns that require revisions. Some of those concerns result

from a study by Khatiwala et al. that is currently in review with Science Advances. We hope that it will be published soon so that the authors can access it and consider it in their revision. Khatiwala et al. use a data-constrained model of the LGM to decompose the carbon cycle. They show that using the AOU approximation to calculate respired carbon leads to large errors (even the wrong sign) in LGM – PI simulations. This conclusion is supported by previous studies who have demonstrated the errors in the AOU approximation (Russell et al., 2003; Ito et al., 2004; Duteil et al., 2013). For this reason, I would advise not to use it and remove the corresponding parts of the manuscript (e.g. in section 3.2). It is OK to refer to the iron fertilization effect as increasing the efficiency of the biological pump, but not that the LGM biological pump was enhanced. Khatiwala et al. show that the biological pump was not enhanced, but that air-sea disequilibrium was increased, which caused the glacial ocean carbon inventory to be larger. Air-sea disequilibrium was enhanced in the LGM not only for carbon but also for oxygen and radiocarbon. Khatiwala et al. show that in their best fitting model the ideal age of the whole ocean is younger, while the whole ocean c14-age is older due to the increased disequilibrium (or increased preformed c14-age). This is relevant for the discussion at the end of section 3 (lines 280-287) and the corresponding parts of the abstract (lines 22-24). Thus, ideal age and c14-age cannot be compared and there may not be a discrepancy here between modeled younger ideal age and older (observed) c14-age. I also think that one quantitative oxygen reconstruction from the Southern Ocean alone (Gottschalk et al. 2016) is not enough to indicate that the model is wrong. Reconstructions have errors and therefore I would not overemphasize this apparent discrepancy.

Another concern is the discussion of nutrient inventory changes. Somes et al. (2017) have considered this and shown that existing nitrogen isotope data provide no constraints on this effect. I'm also not aware of other observations supporting it (including evidence provided in this manuscript). For this reason, I think this effect remains unconstrained by observations and thus highly uncertain. I'd encourage the authors to reflect this uncertainty more in their discussion of this effect and to cite the above paper, which has also examined its effects on oxygen.

The authors claim that their model fits reconstructions of export production by Kohfeld et al. (2005), which show not much change in the Pacific sector of the Southern Ocean. However, there are some newer data from that region by Studer et al. (2015) and Wang et al. (2017) that indicate increased nutrient utilization there as well. This suggests that the model underestimates iron fertilization in the Pacific sector of the Southern Ocean. In any case, given the uncertainties in existing paleo data and iron models and solubility of iron, it is not fair to say that the upper limit of iron fertilization is 20 ppm as claimed here in lines 214-215. Khatiwala et al. suggest an iron effect of 35 ppm. Here I also disagree with Fortunat's suggestion to mention the CO2 limit in the abstract. I don't think it is a robust result. However, the idea that the effect of iron fertilization is limited and that increasing fluxes will have a smaller effect at high fluxes than at low fluxes is robust and agrees with previous results (Muglia et al., 2018). The latter paper suggests this limitation is due to increased scavenging rather than reduced regions of iron limitation. Both seem plausible explanations.

Minor comments: Line 16-17: I suggest to remove "(e.g. more sluggish ocean circulation)" because no such attribution was done in the paper. Khatiwala et al. suggest no CO2 effect from ocean circulation changes.

Line 17-18: I suggest to remove "enhanced efficiency of the biological pump" here for the above mentioned reasons.

Line 21: this sentence is awkward. I suggest to rephrase to "glacial deep water was a more severe environment for . . . than the modern ocean."

Lines 24, 26: again, I'd suggest to rephrase to avoid using the term "biological pump" because it has not been quantified how much CO2 change was due to biological pump changes. Perhaps better to use "iron fertilization and/or global nutrient increase".

Line 31: the biological pump also includes the CaCO3 pump

Lines 50-51: consider including Schmittner and Somes (2015) and Somes et al. (2017)

who have also looked at oxygen

Lines 51-52: Khatiwala et al. have explored oxygen changes in more detail

Line 83: see above comments on "biological pump"

109-110: iron solubility is modified by transport in the atmosphere. This leads to increasing solubility at lower concentrations. This effect has been considered in Muglia et al. (2017; their Fig. 2). This suggests using a constant solubility is not correct. This should be discussed.

116-119: This is about a factor of 10 increase in the 3% experiments. Compare with Muglia et al. (2018) who only have a factor of 4 increase in their best fitting model, which is constrained by d15N and d13C data.

129-130: Muglia et al. (2017) shows the sea level effect to be important.

General comment on section 2: how was the effect of sea level lowering on benthic denitrification treated? Somes et al. (2017) show that this effect reduces N loss in the LGM ocean and leads to a larger N inventory.

165: delete: "because dust deposition flux of the Southern Ocean is underestimated in LGM_dust"

166: delete "in the" and "with iron limitation"

167: delete "in the"

182-184: see above comment on new data from the S. Pacific

199-201: see above comments on biological pump. I doubt that this conclusion is true because of the use of the AOU approximation here, which compromises the results.

239: replace "is the one" with "may be one of the". Or even better remove this whole part due to my above comments.

243: typo: "whehre"

265-266: Schmittner and Somes (2015) and Somes et al. (2017) also get a deep ocean O2 decrease

318: I don't think that's an issue. See comments above.

Khatiwala, S., A. Schmittner, and J. Muglia (in revision), Air-sea disequilibrium enhances ocean carbon storage during glacial periods, Science Advances.

Ito, T., M. J. Follows, and E. A. Boyle (2004), Is AOU a good measure of respiration in the oceans?, Geophys Res Lett, 31(17), doi: 10.1029/2004GL020900.

Duteil, O., W. Koeve, A. Oschlies, D. Bianchi, E. Galbraith, I. Kriest, and R. Matear (2013), A novel estimate of ocean oxygen utilisation points to a reduced rate of respiration in the ocean interior, Biogeosciences, 10(11), 7723-7738, doi: 10.5194/bg-10-7723-2013.

Russell, J. L., and A. G. Dickson (2003), Variability in oxygen and nutrients in South Pacific Antarctic Intermediate Water, Global Biogeochem Cy, 17(2), doi: 10.1029/2000gb001317.

Somes, C. J., A. Schmittner, J. Muglia, and A. Oschlies (2017), A Three-Dimensional Model of the Marine Nitrogen Cycle during the Last Glacial Maximum Constrained by Sedimentary Isotopes, Frontiers in Marine Science, 4(108), doi: 10.3389/fmars.2017.00108.

Studer, A. S., D. M. Sigman, A. Martínez-García, V. Benz, G. Winckler, G. Kuhn, O. Esper, F. Lamy, S. L. Jaccard, L. Wacker, S. Oleynik, R. Gersonde, and G. H. Haug (2015), Antarctic Zone nutrient conditions during the last two glacial cycles, Paleoceanography, 30(7), 845-862, doi: 10.1002/2014PA002745.

Wang, X. T., D. M. Sigman, M. G. Prokopenko, J. F. Adkins, L. F. Robinson, S. K. Hines, J. Chai, A. S. Studer, A. Martínez-García, T. Chen, and G. H. Haug (2017), Deep-sea coral evidence for lower Southern Ocean surface nitrate concentrations during the last ice age, Proceedings of the National Academy of Sciences, 114(13), 3352, doi:

10.1073/pnas.1615718114.

Muglia, J., L. C. Skinner, and A. Schmittner (2018), Weak overturning circulation and high Southern Ocean nutrient utilization maximized glacial ocean carbon, Earth Planet Sc Lett, 496, 47-56, doi: 10.1016/j.epsl.2018.05.038.

Schmittner, A., and C. J. Somes (2016), Complementary constraints from carbon (13C) and nitrogen (15N) isotopes on the glacial ocean's soft-tissue biological pump, Paleoceanography, 669–693, doi: 10.1002/2015PA002905.

---

## Author Comment (AC1) · 3 May 2019

Response to Fortunat Joos (Referee)

(Our response **highlighted gray**.)

**General comment:**

Yamamoto and colleagues present an interesting analysis of glacial change in atmospheric CO2 and marine oxygen. The authors investigate, using a range of factorial analyses, the impacts of glaciogenic iron input and an increased nutrient inventory in the glacial ocean. They apply an offline biogeochemical model for Last Glacial Maximum (LGM) and preindustrial (PI) conditions. They simulate an upper limit for the CO2 decrease due to iron fertilization of 20 ppm and a similar decrease due to an increase in whole ocean nutrient inventory. They present a novel model-proxy comparison for PI-LGM changes in O2. The results suggest a role of iron fertilization and changes in nutrient inventory for low glacial CO2 and for the reconstructed oxygen changes. The manuscript is concise and well written. Figures and tables are illustrative and support the conclusions.

I recommend publication of the manuscript after minor revision.

**Response: We appreciate the positive recommendation and helpful comments from Professor Fortunat Joos. We reply to each specific comment below.**

Comment #1

I find it interesting that the upper limit for iron fertilization is 20 ppm (p10, l215). I would appreciate if this finding is lifted to the abstract.

**Response: Thank you very much for this positive comment. We also think that this result is interesting. However, as mentioned by another referee (Professor Andreas Schmittner), there remains a possibility that this upper limit for iron fertilization is not a robust result because present iron models have a large uncertainty. Thus, we will not mention the upper limit of iron fertilization in the abstract. To obtain a deeper understanding of the impact of iron fertilization on glacial $CO_2$ decrease, the variability of upper limit among iron models should be investigated in the future study.**

Comment #2

Figure 8 shows results from WOA2009 and simulated anomalies. Results for the model for the modern ocean should be displayed as well. This would permit the reader to assess the quality of the simulated O2 field.

**Response: According to the reviewer's comment, we will add the simulated $O_2$ distribution for the modern ocean to Figure 8. The following figure and caption are the revised version of Figure 8.**

[Figure]

**Figure 8. Expansion of hypoxic waters. Horizontal distribution of thickness of hypoxic waters ($[O_2]$ <80 mmol m$^{-3}$) for (a) present, (b) PI and (c) LGM_all. (d) Vertical distribution of hypoxic waters for the present (black solid), PI (black dashed), LGM_clim (red), and LGM_all (blue). Because present coarse resolution models have difficulties in reproducing low oxygen concentration for the present day (Bopp et al., 2013), observed values from WOA2009 (Garcia et al., 2010a) are used for the present. For the LGM simulations, we combine the observed values with the modelled changes.**

Comment #3

There are some language problems, e.g. missing articles, and the manuscript would benefit from proof-reading by a native speaker.

**Response: We will ask a native speaker to performed proof-reading of our manuscript.**

Comment #4

There is no discussion on the role of the burial-nutrient feedback and how burial-nutrient feedback may affect the results of this study. On page 10, l221, it is mentioned that CaCO3 compensation is not included. However, this study does also not consider how changes in iron fertilization affect the balance between weathering and burial of organic matter. This also applies to some extent to the experiment with the increase in whole ocean nutrient inventory.

Several studies point to the potentially important role of the ocean/sediment/lithosphere fluxes of organic matter and how the associated burial-nutrient feedback modifies the magnitude and time scales of the response in CO2 and other tracers to changes in the marine biological cycles (Wallmann et al., 2016;Roth et al., 2014;Jeltsch-Thömmes et al., 2018). (Tschumi et al., 2011), for example, quantify the implication of ocean-sediment-lithosphere coupling for an

experiment where the ocean P inventory is increased. (Menviel et al., 2012) present results from factorial experiments with altered iron fertilization/dust input and altered P inventory plus variation in other drivers from transient glacial-interglacial simulations. I suggest that this caveat is addressed on page 10 and perhaps also in the discussion section.

**Response: Thank you for your useful suggestion. We will add the discussion about the role of the burial-nutrient feedback to page 10, L221-224 as follow.**

**"Note that changes in sedimentation process (i.e., carbonate compensation and burial-nutrient feedback) are not considered in our simulations. The simulated increase in the bottom water DIC (Fig. 4) would enhance dissolution of calcium carbonate in the sediments and thereby increase ocean alkalinity, leading to further $CO_2$ decline (Bouttes et al., 2011; Brovkin et al., 2012; Kobayashi et al., 2018). Long-term balance between burial of organic material and nutrient input through weathering is also potentially important for the response in atmospheric $CO_2$ and related tracers to changes in the ocean biological cycles (Roth et al., 2014; Wallmann et al., 2016). For example, Tschumi et al (2011) show that the nutrient-burial feedback significantly amplifies the effect of increase in $PO_4$ inventory on glacial $CO_2$ decrease. Menviel et al (2012) quantify the implication of ocean-sediment-lithosphere coupling for factorial experiments with altered iron fertilization and altered PO4 inventory from transient glacial-interglacial simulations. Considering that EP increase due to iron fertilization and nutrient increase is smaller in our simulations than in previous studies (Tschumi et al., 2011; Menviel et al., 2012), the effect of burial-nutrient feedback on the reduction of glacial $CO_2$ may be smaller than previous estimation."**

Minor and technical comments
**1**
P1, line 11, p3, l46: ".. due to sea surface cooling" What matters is in my opinion the cooling of the whole ocean, including the ocean interior. Please modify the wording

**Response: In the revised text "due to sea surface cooling" will be changed to "due to seawater cooling".**

**2**
P1, l16-18: This sentence is not so clear. The circulation changes itself likely induce a change in the efficiency of the biological pump (Volk and Hoffert, 1985) as may also be seen when looking at preformed/remineralized nutrients or AOU. I think it should rather read "whereas the other half is driven by iron fertilization and an increase in whole ocean P inventory" or similar.

**Response: We agree fully with the referee on this point. We will revise this sentence as follow.**

**"Sensitivity experiments reveal that physical changes contribute to only half of all glacial deep deoxygenation, whereas the other half is driven by iron fertilization and an increase in whole ocean nutrient inventory"**

**3**

P5, l90: Is convection included in the offline model and how is this done?

**Response: Yes, effects of convection are included in offline model by enhancing the value of the vertical diffusivity where the convection takes place.**

**4**

P9, l192, You may also refer to (Menviel et al., 2012)

**5**

P8, l182: missing word: "shortwave radiation"

**6**

P10, l207: you may include here EMICs results (e.g. (Muglia et al., 2017; Parekh et al., 2008; Menviel et al., 2012;Heinze et al., 2016).

**Response: As for these three comments, we will add the suggested reference and missing word to the revised manuscript. We would like to thank the reviewer for the attention to detail.**

---

## Author Comment (AC2) · 3 May 2019

Response to Andreas Schmittner (Referee)

(Our response **highlighted gray**.)

**General comment:**

Yamamoto and co-workers present a nice modeling study of glacial ocean oxygen and carbon changes. The manuscript is well written (except for a few typos) and nicely illustrated. I think the main new finding is that glaciological iron sources from Patagonia are particularly important for lowering atmospheric CO2. Although similar suggestions have been made previously with simpler models (e.g. Brovkin et al., 2007) this study is the first to my knowledge that cleanly separates glaciological from other (desert) dust sources.

However, I have a few concerns that require revisions. Some of those concerns result from a study by Khatiwala et al. that is currently in review with Science Advances. We hope that it will be published soon so that the authors can access it and consider it in their revision.

**Response: We are grateful to Professor Andreas Schmittner for careful review and useful comments. The reviewer's comments are helpful for us to improve our manuscript. Referring to the comments, we will carefully revise the manuscript. The specific replies are as follows.**

Comment #1

Khatiwala et al. use a data-constrained model of the LGM to decompose the carbon cycle. They show that using the AOU approximation to calculate respired carbon leads to large errors (even the wrong sign) in LGM – PI simulations. This conclusion is supported by previous studies who have demonstrated the errors in the AOU approximation (Russell et al., 2003; Ito et al., 2004; Duteil et al., 2013). For this reason, I would advise not to use it and remove the corresponding parts of the manuscript (e.g. in section 3.2).

**Response: We were not aware of these previous studies and agree that AOU contains errors. In the revised manuscript we will not remove section 3.2, but add the following annotation after line 202. We will refer Khatiwala et al. if their manuscript will be accepted during the revision of our manuscript.**

**"It is important to note that the AOU is different from true oxygen utilization due to air-sea disequilibrium which is on the order of 20 mmol m$^{-3}$ in deep-water formation regions (Russell and Dickson, 2003; Duteil et al., 2013). Changes in surface ocean disequilibrium between PI and LGM simulations might lead to large errors in AOU changes."**

Comment #2

It is OK to refer to the iron fertilization effect as increasing the efficiency of the biological pump, but not that the LGM biological pump was enhanced. Khatiwala et al. show that the biological pump was not enhanced, but that air-sea disequilibrium was increased, which caused the glacial ocean carbon inventory to be larger.

**Response: According to the reviewer's comment, "enhanced biological pump" and/or "biological pump was enhanced" will be removed in the revised manuscript.**

Comment #3

Air-sea disequilibrium was enhanced in the LGM not only for carbon but also for oxygen and radiocarbon. Khatiwala et al. show that in their best fitting model the ideal age of the whole ocean is younger, while the whole ocean c14-age is older due to the increased disequilibrium (or increased preformed c14-age). This is relevant for the discussion at the end of section 3 (lines 280-287) and the corresponding parts of the abstract (lines 22-24). Thus, ideal age and c14-age cannot be compared and there may not be a discrepancy here between modeled younger ideal age and older (observed) c14-age. I also think that one quantitative oxygen reconstruction from the Southern Ocean alone (Gottschalk et al. 2016) is not enough to indicate that the model is wrong. Reconstructions have errors and therefore I would not overemphasize this apparent discrepancy.

**Response: Thank you for sharing the manuscript. We will add the discussion about the effect of air-sea disequilibrium on ideal age and c14-age if Khatiwala et al. will be accepted during the revision of our manuscript.**

Comment #4

Another concern is the discussion of nutrient inventory changes. Somes et al. (2017) have considered this and shown that existing nitrogen isotope data provide no constraints on this effect. I'm also not aware of other observations supporting it (including evidence provided in this manuscript). For this reason, I think this effect remains unconstrained by observations and thus highly uncertain. I'd encourage the authors to reflect this uncertainty more in their discussion of this effect and to cite the above paper, which has also examined its effects on oxygen.

**Response: Thank you for your useful suggestion. As reviewer said, the discussion of nutrient inventory changes is necessary for our manuscript. We will add the following discussion to the revised manuscript.**

**"The changes in nutrient inventory during the LGM have large uncertainties. Previous studies estimate that the oceanic $PO_4$ and $NO_3$ inventories could have been 15–40% (Tamburini and Föllmi, 2009; Wallman et al., 2016) and 10–100% (Deutsch et al., 2004; Eugster et al., 2013; Somes et al., 2017) greater during glacial than interglacial periods, respectively. Moreover, Somes et al (2017) shows that sedimentary $\delta^{15}N$ records provide no constraints on this effect. Future simulations should test the biogeochemical sensitivity to changes in nutrient inventory."**

Comment #5

The authors claim that their model fits reconstructions of export production by Kohfeld et al. (2005), which show not much change in the Pacific sector of the Southern Ocean. However, there are some newer data from that region by Studer et al. (2015) and Wang et al. (2017) that indicate increased nutrient utilization there as well. This suggests

that the model underestimates iron fertilization in the Pacific sector of the Southern Ocean.

**Response: Thank you for information on new data. In addition to suggested references, we also found Kohfeld et al (2013) which include many reconstructions of export production in the Pacific sector of the Southern Ocean. We will add these data to figure 3 and also revise lines 182-188 as follow.**

**"In the model, EP changes also have an east-west dipole pattern; slight increases of EP are found in the South Pacific Ocean and significant EP increases occur in the South Atlantic and Indian Oceans. We found that this pattern is attributed to iron fertilization by glaciogenic dust. Glaciogenic dust derived from Patagonian glaciers is transported to the South Atlantic and Indian Oceans by the southern westerly wind, but is unable to reach the South Pacific (Fig. S2). Proxy data show no clear east-west dipole pattern, suggesting that the model underestimates iron fertilization in the Pacific sector of the Southern Ocean. However, proxy data in the South Pacific are still sparse and quantitative comparison of EP changes between South Atlantic and South Pacific is limited. Therefore, further proxy data in the South Pacific is required for a comprehensive understanding of the glacial EP changes and iron fertilization."**

[Figure]

**Figure 3. Model-proxy comparison of EP change from the PI to LGM. EP difference from the PI for (a) LGM_clim, (b) LGM_glac3%, and (c) LGM_all. Circles show proxy data (Kohfeld et al., 2013). Solid (dotted) lines refer to the glacial sea ice fraction of 0.1 in August (February). (d) Zonal mean changes in surface EP from the PI for LGM_clim (black), LGM_glac3% (red), and LGM_all (blue).**

Comment #6

In any case, given the uncertainties in existing paleo data and iron models and solubility of iron, it is not fair to say that the upper limit of iron fertilization is 20 ppm as claimed here in lines 214-215. Khatiwala et al. suggest an iron effect of 35 ppm. Here I also disagree with Fortunat's suggestion to mention the CO2 limit in the abstract. I don't think it is a robust result. However, the idea that the effect of iron fertilization is limited and that increasing fluxes will have a smaller effect at high fluxes than at low fluxes is robust and agrees with previous results (Muglia et al., 2018). The latter paper suggests this limitation is due to increased scavenging rather than reduced regions of iron limitation. Both seem plausible explanations.

**Response: Thank you for this discussion. We will add the following discussion about the uncertainty of upper limit of iron fertilization to the revised manuscript.**

**"The simulated upper limit of $CO_2$ reduction due to iron fertilization would not be a robust result because present iron models have large uncertainty. While Parekh et al (2008) show upper limit of 10 ppm, other simulations show $CO_2$ decrease by more than 20 ppm (Oka et al., 2011; Muglia et al., 2017). To obtain a deeper understanding of the impact of iron fertilization on glacial $CO_2$ decrease, the variability of upper limit among iron models should be investigated in the future study."**

Minor and technical comments

**1**

Line 16-17: I suggest to remove "(e.g. more sluggish ocean circulation)" because no such attribution was done in the paper. Khatiwala et al. suggest no CO2 effect from ocean circulation changes.

**Response: According to the reviewer's comment, we will remove this part.**

**2**

Line 17-18: I suggest to remove "enhanced efficiency of the biological pump" here for the above mentioned reasons.

**Response: In the revised text "by enhanced efficiency of the biological pump" will be changed to "by iron fertilization and an increase in whole ocean nutrient inventory".**

**3**

Line 21: this sentence is awkward. I suggest to rephrase to "glacial deep water was a more severe environment for ... than the modern ocean."

**4**

Lines 24, 26: again, I'd suggest to rephrase to avoid using the term "biological pump" because it has not been quantified how much CO2 change was due to biological pump changes. Perhaps better to use "iron fertilization and/or global nutrient increase".

**Response: Thank you for pointing out. Following reviewer's comments, we will correct these two sentences.**

**5**

Line 31: the biological pump also includes the CaCO3 pump

**Response: We will change "the biological pump" to "the soft-tissue biological pump".**

**6**

Lines 50-51: consider including Schmittner and Somes (2015) and Somes et al. (2017)

**Response: Suggested reference will be added in the revised manuscript.**

**7**

Lines 51-52: Khatiwala et al. have explored oxygen changes in more detail

**Response: We will refer the results of Khatiwala et al. if their manuscript will be accepted during the revision of our manuscript.**

**8**

Line 83: see above comments on "biological pump"

**Response: "enhanced efficiency of biological pump associated with" will be removed in the revised manuscript.**

**9**

109-110: iron solubility is modified by transport in the atmosphere. This leads to increasing solubility at lower concentrations. This effect has been considered in Muglia et al. (2017; their Fig. 2). This suggests using a constant solubility is not correct. This should be discussed.

**Response: We agree that a constant solubility is not correct, as was described in lines 121-123. We will add the discussion about iron solubility to the revised manuscript, as follow.**

**"Present observation shows generally lower Fe solubility at higher Fe concentration in aerosols and higher solubility at lower concentration. Thus, assumed constant iron solubility at 2% in all types of dust could lead to overestimation of total DFe flux from different types of Fe-containing aerosols in LGM (Muglia et al., 2017). On the other hand, much higher Fe solubility (1–42% of Fe solubility) is measured for the LGM aerosols in Antarctica (Conway et al., 2015), suggesting that assumed constant iron solubility at 1% for all types of dust could lead to underestimation of DFe flux in LGM."**

**10**

116-119: This is about a factor of 10 increase in the 3% experiments. Compare with Muglia et al. (2018) who only have a factor of 4 increase in their best fitting model, which is constrained by d15N and d13C data.

Response: We will add following sentences to line 118.

"This value is roughly 10 times larger than in PI simulation and is larger than a recent estimation, which suggest that quadrupling of global DFe flux is constrained by model proxy comparison of $\delta^{15}N$ and $\delta^{13}C$ (Muglia et al., 2018)."

**11**

129-130: Muglia et al. (2017) shows the sea level effect to be important.

Response: "Muglia et al (2017) show this effect causes $CO_2$ increase by 15 ppm." will be added after lines 129-130.

**12**

General comment on section 2: how was the effect of sea level lowering on benthic denitrification treated? Somes et al. (2017) show that this effect reduces N loss in the LGM ocean and leads to a larger N inventory.

Response: Benthic denitrification is not considered in our model. We will add this information to line 150, as follow.

"In our simulations, changes in benthic denitrification is not considered. Somes et al (2017) show that decrease in benthic denitrification due to sea level drop reduces $NO_3$ loss and thus leads to a lager $NO_3$ inventory in the LGM ocean."

**13**

165: delete: "because dust deposition flux of the Southern Ocean is underestimated in LGM_dust"

**14**

166: delete "in the" and "with iron limitation"

**15**

167: delete "in the"

Response: According to reviewer's comments, we will remove these three parts.

**16**

182-184: see above comment on new data from the S. Pacific

Response: As mentioned above, we will revise lines 182-188 and compare our model with new data of the Southern Pacific.

**17**

199-201: see above comments on biological pump. I doubt that this conclusion is true because of the use of the AOU approximation here, which compromises the results.

Response: As mentioned above, we will add the annotation about errors of AOU approximation.

**18**

239: replace "is the one" with "may be one of the". Or even better remove this whole part due to my above comments.

Response: "is the one" will be changed to "may be one of the".

**19**

243: typo: "whehre"

Response: Thank you for pointing out. We will fix typo in the revised manuscript.

References:

Kohfeld, K. E., Graham, R. M., de Boer, A. M., Sime, L. C., Wolff, E. W., Le Quéré, C., and Bopp, L.: Southern Hemisphere westerly wind changes during the last glacial maximum: Paleo-data synthesis, Quat. Sci. Rev., 68, 76–95, 2013.

Parekh, P., Joos, F., and Muller, S. A.: A modeling assessment of the interplay between aeolian iron fluxes and ironbinding ligands in controlling carbon dioxide fluctuations during Antarctic warm events, Paleoceanography, 23, Pa4202, doi:10.1029/2007pa001531, 2008.

---

## Author Response (AR1)

Dear Editor

We are grateful to you and two reviewers for the helpful comments on the previous version of our manuscript. We have addressed all the comments made by two reviewers, as follows. Generally, we revised the manuscript according to reviewer's comments. This is because the comments of all reviewers are very useful for improving our manuscript and strengthening the interpretation of our model results. Native speaker has performed proofreading of our manuscript and corrected errors and inappropriate expression in English sentences. We hope that the revised version of our paper is now suitable for publication in Climate of the Past.

Response to Fortunat Joos (Referee)

(Our response **highlighted gray**.)

**General comment:**

Yamamoto and colleagues present an interesting analysis of glacial change in atmospheric CO2 and marine oxygen. The authors investigate, using a range of factorial analyses, the impacts of glaciogenic iron input and an increased nutrient inventory in the glacial ocean. They apply an offline biogeochemical model for Last Glacial Maximum (LGM) and preindustrial (PI) conditions. They simulate an upper limit for the CO2 decrease due to iron fertilization of 20 ppm and a similar decrease due to an increase in whole ocean nutrient inventory. They present a novel model-proxy comparison for PI-LGM changes in O2. The results suggest a role of iron fertilization and changes in nutrient inventory for low glacial CO2 and for the reconstructed oxygen changes. The manuscript is concise and well written. Figures and tables are illustrative and support the conclusions.

I recommend publication of the manuscript after minor revision.

**Response: We appreciate the positive recommendation and helpful comments from Professor Fortunat Joos. We reply to each specific comment below.**

Comment #1

I find it interesting that the upper limit for iron fertilization is 20 ppm (p10, l215). I would appreciate if this finding is lifted to the abstract.

**Response: Thank you very much for this positive comment. We also think that this result is interesting. However, as mentioned by another referee (Professor Andreas Schmittner), there remains a possibility that**

**this upper limit for iron fertilization is not a robust result because present iron models have a large uncertainty. Thus, we do not mention the upper limit of iron fertilization in the abstract. To obtain a deeper understanding of the impact of iron fertilization on glacial $CO_2$ decrease, the variability of upper limit among iron models should be investigated in the future study.**

Comment #2
Figure 8 shows results from WOA2009 and simulated anomalies. Results for the model for the modern ocean should be displayed as well. This would permit the reader to assess the quality of the simulated O2 field.

**Response: According to the reviewer's comment, we added the simulated $O_2$ distribution for the modern ocean to Figure 8. The following figure and caption are the revised version of Figure 8.**

[Figure]

**Figure 8. Hypoxic waters expansion. Horizontal distribution of thickness of the hypoxic waters ($[O_2]$ <80 mmol m$^{-3}$) for the (a) present, (b) PI, and (c) LGM_all. (d) Vertical distribution of hypoxic waters for the present (black solid), PI (black dashed), LGM_clim (red), and LGM_all (blue). Because current coarse**

resolution models have difficulties reproducing low oxygen concentration for the present day (Bopp et al., 2013), observed values from WOA2009 (Garcia et al., 2010a) were used for the present. For the LGM simulations, we combined the observed values with the modelled changes.

Comment #3

There are some language problems, e.g. missing articles, and the manuscript would benefit from proof-reading by a native speaker.

Response: A native speaker have performed proof-reading of our manuscript.

Comment #4

There is no discussion on the role of the burial-nutrient feedback and how burial-nutrient feedback may affect the results of this study. On page 10, l221, it is mentioned that CaCO3 compensation is not included. However, this study does also not consider how changes in iron fertilization affect the balance between weathering and burial of organic matter. This also applies to some extent to the experiment with the increase in whole ocean nutrient inventory.

Several studies point to the potentially important role of the ocean/sediment/lithosphere fluxes of organic matter and how the associated burial-nutrient feedback modifies the magnitude and time scales of the response in CO2 and other tracers to changes in the marine biological cycles (Wallmann et al., 2016;Roth et al., 2014;Jeltsch-Thömmes et al., 2018). (Tschumi et al., 2011), for example, quantify the implication of ocean-sediment-lithosphere coupling for an experiment where the ocean P inventory is increased. (Menviel et al., 2012) present results from factorial experiments with altered iron fertilization/dust input and altered P inventory plus variation in other drivers from transient glacial-interglacial simulations. I suggest that this caveat is addressed on page 10 and perhaps also in the discussion section.

Response: Thank you for your useful suggestion. We added the following discussion about the role of the burial-nutrient feedback to page11, L246-257 in the revised manuscript.

"Note that changes in the sedimentation process (i.e. carbonate compensation and burial-nutrient feedback) are not considered in our simulation. The simulated increase in the bottom water DIC (Fig. 4) would enhance calcium carbonate dissolution in the sediments and thereby increase ocean alkalinity, leading to a further $CO_2$ decrease (Bouttes et al., 2011; Brovkin et al., 2012; Kobayashi et al., 2018). The long-term balance between the burial of organic material and nutrient input through weathering is also potentially important for the response in atmospheric $CO_2$ and related tracers to changes in ocean biological cycles (Roth et al., 2014; Wallmann et al., 2016). For example, Tschumi et al. (2011) show that the nutrient-burial feedback significantly amplifies the effect of an increase in the $PO_4$ inventory on the glacial $CO_2$ decrease. Menviel et al. (2012) quantified the implication of ocean-sediment-lithosphere coupling for factorial experiments with an altered iron fertilization and altered $PO_4$ inventory from transient glacial-interglacial simulations. Considering that EP increases due to iron fertilization and the nutrient increase is smaller in our simulations than that in previous studies (Tschumi et al., 2011; Menviel et al., 2012), the effect of burial-nutrient feedback on the glacial $CO_2$ reduction may be smaller than previously estimated."

Minor and technical comments
**1**
P1, line 11, p3, l46: ".. due to sea surface cooling" What matters is in my opinion the cooling of the whole ocean, including the ocean interior. Please modify the wording

Response: "due to sea surface cooling" was changed to "due to seawater cooling" in page1 L10-11 in the revised manuscript.

**2**
P1, l16-18: This sentence is not so clear. The circulation changes itself likely induce a change in the efficiency of the biological pump (Volk and Hoffert, 1985) as may also be seen when looking at preformed/remineralized nutrients or AOU. I think it should rather read "whereas the other half is driven by iron fertilization and an increase in whole ocean P inventory" or similar.

Response: We agree fully with the referee on this point. We revised this sentence as follow (page1 L16-18).
 "Sensitivity experiments show that physical changes contribute to only one-half of all glacial deep deoxygenation whereas the other one-half is driven by iron fertilization and an increase in the whole ocean nutrient inventory."

**3**

P5, l90: Is convection included in the offline model and how is this done?

Response: Yes, effects of convection are included in offline model by enhancing the value of the vertical diffusivity where the convection takes place.

**4**

P9, l192, You may also refer to (Menviel et al., 2012)

**5**

P8, l182: missing word: "shortwave radiation"

**6**

P10, l207: you may include here EMICs results (e.g. (Muglia et al., 2017; Parekh et al., 2008; Menviel et al., 2012;Heinze et al., 2016).

Response: As for these three comments, we added the suggested reference and missing word to the revised manuscript (page10 207, page9 L195, page10 L226-228). We would like to thank the reviewer for the attention to detail.

Response to Andreas Schmittner (Referee)
(Our response highlighted gray.)

**General comment:**

Yamamoto and co-workers present a nice modeling study of glacial ocean oxygen and carbon changes. The manuscript is well written (except for a few typos) and nicely illustrated. I think the main new finding is that glaciological iron sources from Patagonia are particularly important for lowering atmospheric CO2. Although similar suggestions have been made previously with simpler models (e.g. Brovkin et al., 2007) this study is the first to my knowledge that cleanly separates glaciological from other (desert) dust sources.

However, I have a few concerns that require revisions. Some of those concerns result from a study by Khatiwala et al. that is currently in review with Science Advances. We hope that it will be published soon so that the authors can access it and consider it in their revision.

**Response: We are grateful to Professor Andreas Schmittner for careful review and useful comments. The reviewer's comments are helpful for us to improve our manuscript. Referring to the comments, we carefully revised the manuscript. The specific replies are as follows.**

Comment #1
Khatiwala et al. use a data-constrained model of the LGM to decompose the carbon cycle. They show that using the AOU approximation to calculate respired carbon leads to large errors (even the wrong sign) in LGM – PI simulations. This conclusion is supported by previous studies who have demonstrated the errors in the AOU approximation (Russell et al., 2003; Ito et al., 2004; Duteil et al., 2013). For this reason, I would advise not to use it and remove the corresponding parts of the manuscript (e.g. in section 3.2).

**Response: We were not aware of these previous studies and agree that AOU contains errors. In the revised manuscript we did not remove section 3.2, but add the following annotation to page10 L218-221.**

**Notably, the AOU is different from true oxygen utilization due to the air-sea disequilibrium which is on the order of 20 mmol m$^{-3}$ in deep-water formation regions (Russell and Dickson, 2003; Duteil et al., 2013). Changes in surface ocean disequilibrium between the PI and LGM simulations might lead to large errors in the AOU changes.**

Comment #2
It is OK to refer to the iron fertilization effect as increasing the efficiency of the biological pump, but not that the LGM biological pump was enhanced. Khatiwala et al. show that the biological pump was not enhanced, but that air-sea disequilibrium was increased, which caused the glacial ocean carbon inventory to be larger.

**Response: According to the reviewer's comment, "enhanced biological pump" and "biological pump was enhanced" are removed in the revised manuscript.**

Comment #3

Air-sea disequilibrium was enhanced in the LGM not only for carbon but also for oxygen and radiocarbon. Khatiwala et al. show that in their best fitting model the ideal age of the whole ocean is younger, while the whole ocean c14-age is older due to the increased disequilibrium (or increased preformed c14-age). This is relevant for the discussion at the end of section 3 (lines 280-287) and the corresponding parts of the abstract (lines 22-24). Thus, ideal age and c14-age cannot be compared and there may not be a discrepancy here between modeled younger ideal age and older (observed) c14-age. I also think that one quantitative oxygen reconstruction from the Southern Ocean alone (Gottschalk et al. 2016) is not enough to indicate that the model is wrong. Reconstructions have errors and therefore I would not overemphasize this apparent discrepancy.

**Response: Thank you for sharing the manuscript. We did not add the discussion about the effect of air-sea disequilibrium on ideal age and c14-age because, at the moment, Khatiwala et al. has not been formally accepted.**

Comment #4

Another concern is the discussion of nutrient inventory changes. Somes et al. (2017) have considered this and shown that existing nitrogen isotope data provide no constraints on this effect. I'm also not aware of other observations supporting it (including evidence provided in this manuscript). For this reason, I think this effect remains unconstrained by observations and thus highly uncertain. I'd encourage the authors to reflect this uncertainty more in their discussion of this effect and to cite the above paper, which has also examined its effects on oxygen.

**Response: Thank you for your useful suggestion. As reviewer said, the discussion of nutrient inventory changes is necessary for our manuscript. We added the following discussion to the revised manuscript (page15 L335-339).**

**"The changes in nutrient inventory during the LGM have large uncertainties. Previous studies estimate that the oceanic $PO_4$ and $NO_3$ inventories could have been 15–40% (Tamburini and Föllmi, 2009; Wallmann et al., 2016) and 10-100% (Deutsch et al., 2004; Eugster et al., 2013; Somes et al., 2017) greater during glacial compared to interglacial periods, respectively. Moreover, Somes et al. (2017) shows that sedimentary $\delta^{15}N$**

**records provide no constrain on this effect. Future simulations should test the biogeochemical sensitivity to nutrient inventory changes."**

Comment #5

The authors claim that their model fits reconstructions of export production by Kohfeld et al. (2005), which show not much change in the Pacific sector of the Southern Ocean. However, there are some newer data from that region by Studer et al. (2015) and Wang et al. (2017) that indicate increased nutrient utilization there as well. This suggests that the model underestimates iron fertilization in the Pacific sector of the Southern Ocean.

**Response: Thank you for information on new data. In addition to suggested references, we also found Kohfeld et al (2013) which include many reconstructions of export production in the Pacific sector of the Southern Ocean. We added these data to figure 3 and also revised page9 L195-203 as follow.**

**"In the model, the EP changes also have an east-west dipole pattern; slight EP increases are found in the South Pacific Ocean and significant EP increases occur in the South Atlantic and Indian oceans. We found that this pattern is attributed to iron fertilization by glaciogenic dust. Glaciogenic dust derived from Patagonian glaciers is transported to the South Atlantic and Indian oceans by the southern westerly wind; however, it is unable to reach the South Pacific (Fig. S3). Proxy data show no clear east-west dipole pattern, suggesting that the model underestimates iron fertilization in the Pacific sector of the Southern Ocean. However, proxy data in the South Pacific remain sparse and a quantitative comparison of EP changes between the South Atlantic and South Pacific is limited. Therefore, further proxy data in the South Pacific is required for a comprehensive understanding of the glacial EP changes and iron fertilization."**

[Figure]

**Figure 3. Model-proxy comparison of EP change from the PI to LGM. The EP difference from the PI for (a) LGM_clim, (b) LGM_glac3%, and (c) LGM_all. Circles show proxy data (Kohfeld et al., 2013). Solid (dotted) lines refer to the glacial sea ice fraction of 0.1 during August (February). (d) Zonal mean changes in the surface EP from the PI for LGM_clim (black), LGM_glac3% (red), and LGM_all (blue).**

Comment #6

In any case, given the uncertainties in existing paleo data and iron models and solubility of iron, it is not fair to say that the upper limit of iron fertilization is 20 ppm as claimed here in lines 214-215. Khatiwala et al. suggest an iron effect of 35 ppm. Here I also disagree with Fortunat's suggestion to mention the CO2 limit in the abstract. I don't think it is a robust result. However, the idea that the effect of iron fertilization is limited and that increasing fluxes will have a smaller effect at high fluxes than at low fluxes is robust and agrees with previous results (Muglia et al., 2018). The latter paper suggests this limitation is due to increased scavenging rather than reduced regions of iron limitation. Both seem plausible explanations.

**Response: Thank you for this discussion. We added the following discussion about the uncertainty of upper limit of iron fertilization to the revised manuscript (page11 L238-241).**

**"The simulated upper limit of $CO_2$ reduction resulting from iron fertilization is not a robust result because present iron models have large uncertainty. While Parekh et al. (2008) show an upper limit of 10 ppm, other simulations show $CO_2$ decrease by greater than 20 ppm (Oka et al., 2011; Muglia et al., 2017). To obtain a better understanding of the impact of iron fertilization on glacial $CO_2$ decrease, the variability of the upper limit among iron models should be investigated in a future study."**

Minor and technical comments

**1**

Line 16-17: I suggest to remove "(e.g. more sluggish ocean circulation)" because no such attribution was done in the paper. Khatiwala et al. suggest no CO2 effect from ocean circulation changes.

**Response: According to the reviewer's comment, we removed this part.**

**2**

Line 17-18: I suggest to remove "enhanced efficiency of the biological pump" here for the above mentioned reasons.

**Response: In the revised text "by enhanced efficiency of the biological pump" was changed to "by iron fertilization and an increase in the whole ocean nutrient inventory". (page1 L17-18)**

**3**

Line 21: this sentence is awkward. I suggest to rephrase to "glacial deep water was a more severe environment for ... than the modern ocean."

**4**

Lines 24, 26: again, I'd suggest to rephrase to avoid using the term "biological pump" because it has not been quantified how much CO2 change was due to biological pump changes. Perhaps better to use "iron fertilization and/or global nutrient increase".

**Response: Thank you for pointing out. Following reviewer's comments, we corrected these two sentences. (page1 L21-22, page1 L24-25)**

**5**

Line 31: the biological pump also includes the CaCO3 pump

**Response: We changed "the biological pump" to "the soft-tissue biological pump". (page3 L32)**

**6**

Lines 50-51: consider including Schmittner and Somes (2015) and Somes et al. (2017)

**Response: Suggested references are added in the revised manuscript (page3 L52).**

**7**

Lines 51-52: Khatiwala et al. have explored oxygen changes in more detail

**Response: We did not refer Khatiwala et al. because, at the moment, it has not been formally accepted.**

**8**

Line 83: see above comments on "biological pump"

**Response: "enhanced efficiency of biological pump associated with" will be removed in the revised manuscript (page5 L84).**

**9**

109-110: iron solubility is modified by transport in the atmosphere. This leads to increasing solubility at lower concentrations. This effect has been considered in Muglia et al. (2017; their Fig. 2). This suggests using a constant solubility is not correct. This should be discussed.

**Response: We agree that a constant solubility is not correct. We added the discussion about iron solubility to the revised manuscript (page6 L123-130), as follow.**

"Present observation generally shows a lower Fe solubility at a higher Fe concentration in aerosols and a higher solubility at a lower concentration (Fig. S1). A wider range of aerosol Fe solubility (from 0.2% to 48%) has been derived from observations over the SO, but different types of Fe-containing minerals such as pyrogenic Fe oxides can be considered to achieve high Fe solubilities (Ito et al., 2019). Thus, an assumed constant iron solubility of 2% in all types of dust could lead to overestimation of a total DFe flux from different types of Fe-containing aerosols during the LGM (Muglia et al., 2017). However, a much higher Fe solubility (1–42% of Fe solubility) as derived from observations for the LGM aerosols in Antarctica has suggested that an assumed constant iron solubility of 1–2% for all types of dust could lead to a DFe flux underestimation during the LGM (Conway et al., 2015)."

**10**
116-119: This is about a factor of 10 increase in the 3% experiments. Compare with Muglia et al. (2018) who only have a factor of 4 increase in their best fitting model, which is constrained by d15N and d13C data.

Response: We added following sentences to page7 L131-133 in the revised manuscript.

"This value is approximately 10 times larger than that of the PI simulation and is larger than a recent estimation, suggesting that a quadrupling of the global DFe flux is constrained by a model-proxy comparison of $\delta^{15}N$ and $\delta^{13}C$ (Muglia et al., 2018)."

**11**
129-130: Muglia et al. (2017) shows the sea level effect to be important.

Response: "Muglia et al (2017) showed this effect causes $CO_2$ increase of 15 ppm." is added to page 7 L141-142.

**12**
General comment on section 2: how was the effect of sea level lowering on benthic denitrification treated? Somes et al. (2017) show that this effect reduces N loss in the LGM ocean and leads to a larger N inventory.

**Response: Benthic denitrification is not considered in our model. We added this information to page8 L162-164, as follow.**

**"In our simulations, changes in benthic denitrification were not considered. Somes et al. (2017) show that a decrease in benthic denitrification because of a sea level drop reduces $NO_3$ loss and thus leads to a larger $NO_3$ inventory in the LGM ocean."**

**13**
165: delete: "because dust deposition flux of the Southern Ocean is underestimated in LGM_dust"

**14**
166: delete "in the" and "with iron limitation"

**15**
167: delete "in the"

**Response: According to reviewer's comments, we removed these three parts (page9 L179, page9 L180).**

**16**
182-184: see above comment on new data from the S. Pacific

**Response: As mentioned above, we revised page9 L195-203 and compared our model with new data of the Southern Pacific.**

**17**
199-201: see above comments on biological pump. I doubt that this conclusion is true because of the use of the AOU approximation here, which compromises the results.

**Response: As mentioned above, we added the annotation about errors of AOU approximation (page10 L218-221).**

**18**

239: replace "is the one" with "may be one of the". Or even better remove this whole part due to my above comments.

**Response: "is the one" was changed to "may be one of the" (page12 L271).**

**19**
243: typo: "whehre"

**Response: Thank you for pointing out. We fixed typo in the revised manuscript.**

References:

[revised manuscript text omitted]

---

## Author Response (AR2)

Editor's comments to the Author:

Please take into account the comments below to make a few minor corrections to your revised manuscript before publication.

Please also include the mandatory sections on data availability and author contribution.

We wish to express our appreciation for the editor for the careful comments and suggestions. We modified the manuscript following your suggestions one by one.  The sections on data availability and author contribution are added before the acknowledgements. We refer Menviel et al (2017), which suggest weaker and shallower NADW and weaker AABW at the

LGM from model-data comparison of $\delta^{13}$C and radiocarbon, in the end of Section 3.3. Our replies are highlighted gray.

Abstract: Please use only present tense. For example: L. 8: "We find", L.22: "Our model underestimates"

changed

P17, L. 62-63: "atmospheric CO2 is not directly transferred to the deep ocean" Maybe you can rephrase as a "transfer of carbon from the atmosphere to the deep ocean"

changed following your suggestion

P23, L.27: "a slight EP increase is found"

changed

P24, L. 85: Please add reference to Khatiwala et al. accepted.

added

P25, L. 26: "a glacial CO2 decrease of more than 30 ppm"

changed

P25, L. 31-p26, L. 48: "important for the atmospheric CO2 response (Roth et al., 2014; Wallmann et al., 2016)."

changed as you suggested

P26, L. 52: "EP increases …and…the nutrient increase are smaller"

changed

P28, L.36: "the underestimation of deoxygenation"

changed

P29, L.72: I would suggest to remove "In conclusion, "

removed

P29, L. 73: Please change "responsible for the glacial CO2 decline of greater than 30 ppm" to "could explain a glacial CO2

decline of more than 30 ppm" or "decline greater than 30 ppm"

[revised manuscript text omitted]